

# Assessment of the 11-year solar cycle signals in the middle atmosphere in multiple-model ensemble simulations

Wenjuan Huo[1], Tobias Spiegl[2], Sebastian Wahl[1], Katja Matthes[1], Ulrike Langematz[3], Holger Pohlmann[4], and Jürgen Kröger[4]

[1]GEOMAR Helmholtz Centre for Ocean Research Kiel, 24148 Kiel,Germany
[2]Alfred-Wegener-Institut, Helmholtz-Zentrum für Polar- und Meeresforschung, 27570, Bremerhaven, Germany
[3]Freie Universität Berlin, 12165 Berlin, Germany
[4]Max Planck Institute for Meteorology, 20146 Hamburg, Germany

**Correspondence:** Wenjuan Huo (whuo@geomar.de)

**Abstract.** To better understand possible reasons for the diverse modeling results and large discrepancies of the detected solar fingerprints, we took one step back and assessed the "initial" solar signals in the middle atmosphere based on large ensemble simulations with multiple climate models — FOCI, EMAC, and MPI-ESM-HR. Consistent with previous work, we find that the 11-year solar cycle signals in the short wave heating rate (SWHR) and ozone anomalies are robust and statistically significant in all three models. These initial solar cycle signals in SWHR, ozone, and temperature anomalies are sensitive to the strength of the solar forcing. Correlation coefficients of the solar cycle with the SWHR, ozone, and temperature anomalies linearly increase along with the enhancement of the solar cycle amplitude, and this reliance becomes more complex when the solar cycle amplitude exceeds a certain threshold. In addition, the cold bias in the tropical stratopause of EMAC dampens the subsequent results of the initial solar signal. The warm pole bias in MPI-ESM-HR leads to a weak polar night jet (PNJ), which may limit the top-down propagation of the initial solar signal. Although FOCI simulated a so-called top-down response as revealed in previous studies in a period with large solar cycle amplitudes, its warm bias in the tropical upper stratosphere results in a positive bias in PNJ and can lead to a "reversed" response in some extreme cases. We suggest a careful interpretation of the single model result and further re-examination of the solar signal based on more climate models.

## 1 Introduction

Significant effects of the 11-year solar cycle on the middle-atmospheric temperature and constituents have been found in many observational and model studies in recent decades (e.g., see either Gray et al. (2010) or Ward et al. (2021) for a review). However, the modeled responses of the nitrogen dioxide (Hood and Soukharev, 2006; Wang et al., 2020), ozone (Soukharev and Hood, 2006; Swartz et al., 2012; Hood et al., 2015; Maycock et al., 2018), and stratospheric temperature (Mitchell et al., 2015; Matthes et al., 2017) in the middle atmosphere still show discrepancies among various models and observations. Enhanced ab-



sorption of solar UV radiation by ozone and oxygen in the middle atmosphere, with direct solar heating effect during the solar maximum years, can increase the tropical stratopause temperature. Kodera and Kuroda (2002) first proposed that the increase of the tropical stratopause temperature can strengthen the meridional temperature gradient and lead to an intensified polar night jet (PNJ) during wintertime. Anomalous westerly winds associated with the strengthened PNJ can propagate downward via interactions with the upward planetary waves. This proposed "top-down" mechanism was partly confirmed in subsequent studies with additional observational/reanalysis data (Kuroda et al., 2022) or with the aid of idealized simulations based on climate models (Matthes et al., 2006; Thiéblemont et al., 2015; Mitchell et al., 2015; Drews et al., 2022). However, the timing of the "top-down" propagation for the detected solar signal varies from December to March (Kodera and Kuroda, 2002; Drews et al., 2022; Kuroda et al., 2022) and is modulated by internal climate variability, such as phases of the Quasi-Biennial Oscillation (QBO) in the stratosphere (Labitzke, 2005; Matthes et al., 2010) and the Pacific decadal oscillation at the surface (Guttu et al., 2021).

Large discrepancies between the observed solar imprints and the modeling results (Li et al., 2016; Wang et al., 2020; Scaife et al., 2013; Andrews et al., 2015), as well as the inconsistent responses in climate models (Drews et al., 2022; Chiodo et al., 2019; Spiegl et al., 2023), diminish the robustness of the detected "solar signal" and call into question the proposed "top-down" mechanism and its surface response. The "top-down" mechanism whereby the solar responses in the middle atmosphere trigger the downward coupling processes, was widely used to explain the solar influences on the Northern Hemisphere (NH) winter climate, especially the modulation on the North Atlantic Oscillation (NAO) (Kodera and Kuroda, 2005; Scaife et al., 2013; Andrews et al., 2015; Gray et al., 2016; Thiéblemont et al., 2015; Drews et al., 2022; Kuroda et al., 2022). However, the controversial modeling results, shown in the publications of Drews et al. (2022), Chiodo et al. (2019) and Spiegl et al. (2023), reduce the confidence level of the solar-NAO connection and the underlying mechanism. The uncertainties of the simulated solar responses in the middle atmosphere may partly explain the discrepancy of the solar surface imprints. In this study, we will use multiple-model ensemble simulations to evaluate the "initial" solar signals in the middle atmosphere with a consideration of the model stratospheric biases.

Kunze et al. (2020) quantified uncertainties of the 11-year solar signals in the annual mean shortwave heating rates (SWHR), temperature, and ozone anomalies in the middle atmosphere based on two chemistry-climate models (CCMs)— ECHAM/MESSy Atmospheric Chemistry (EMAC) and Community Earth System Model and Whole Atmosphere Chemistry Climate Model (CESM-WACCM). They found that the uncertainties of the solar responses in the SWHR, temperature, and ozone anomalies in the upper stratosphere–lower mesosphere arise mainly from the used solar spectral irradiance (SSI) dataset, but solar responses in the lower stratosphere also depend on the CCM used. Recent studies demonstrated that uncertainties of the solar-related dynamical responses in the stratosphere and troposphere, as well as the surface responses, are much larger than the initial solar signals in the middle atmosphere (Drews et al., 2022; Spiegl et al., 2023). A large spread in the dynamical responses, even with an opposite sign, has been found among individual ensemble members based on the same model and with identical solar forcing data (Spiegl et al., 2023). Besides, the dynamics and the uncertainty of the model also can influence the detection of solar imprints and lead to large discrepancies in the solar responses among multiple climate models (Kunze et al., 2020). An



evaluation of the 11-year solar signal in multiple-model ensemble simulations will update the information on the robustness and significance of solar imprints in Earth's atmosphere and also be helpful for the development of CCMs.

Although many studies show the importance of the middle atmosphere for the troposphere and surface climate, model representations of all the chemical, physical, and dynamic processes in the middle atmosphere are still a very big challenge (Scaife et al., 2022; Lawrence et al., 2022). Recently, longer predictability timescales of the stratosphere as compared to the

60 troposphere have been identified (Tripathi et al., 2015; Butler et al., 2019; Domeisen et al., 2020a). The stratosphere and its coupling with the troposphere could be a source for the predictability of wintertime surface weather on subseasonal-to-seasonal (S2S) timescales (Hardiman et al., 2011; Jia et al., 2017; Domeisen et al., 2020b; Scaife et al., 2022). During the solar maximum years, the strong solar radiative effects in the upper middle atmosphere (direct heating and absorption of solar UV radiation) could change its thermal and dynamical features and hence influence the planetary wave propagation and reflection

conditions in the lower middle atmosphere (Kodera and Kuroda, 2002; Matthes et al., 2006; Drews et al., 2022; Kuroda et al., 2022). Therefore, the quasi-decadal solar forcing is recognized as a potential source for near-term climate prediction (Scaife et al., 2022; Kushnir et al., 2019). However, climate models and forecast systems often struggle to correctly represent all stratospheric variability and their downward coupling processes (Scaife et al., 2016; Lawrence et al., 2022). The inclusion of the solar response in the middle atmosphere in a forecast system would not only improve the prediction of stratospheric events

but also may help improve surface prediction skills.

Herein, we first assess the initial solar signals in the middle atmosphere (subsection 3.1) as well as the top-down mechanism in multiple model ensemble simulations (subsection 3.2). Furthermore, we investigate the uncertainties of the dynamical responses in climate models with a consideration of the model biases (subsection 3.3). In Section 2 we describe all three models and the setup of the experiments used in this study. In Section 3 we show the results and in Section 4 we provide the conclusions

and discussions.

## 2 Data and Methods

### 2.1 Climate Models

Three climate models are used in this study to assess the 11-year solar cycle signals in the middle atmosphere. They are the Flexible Ocean Climate Infrastructure (hereafter FOCI, Matthes et al. (2020)), the ECHAM/MESSy Atmospheric Chemistry

(hereafter EMAC, Jöckel et al. (2016)), and the Max Planck Institute for Meteorology Earth System Model in high-resolution configuration (hereafter MPI-ESM-HR, Müller et al. (2018)). A brief introduction of the three models is given below.

FOCI is a fully coupled CCM that includes the high-top atmospheric model ECHAM6.3 (Stevens et al., 2013) describing tropospheric and middle-atmospheric processes, coupled to the NEMO3.6 ocean model (Madec, 2016), the JSBACH (Reick et al., 2013) land module as well as the LIM2 (Fichefet and Maqueda, 1997) sea ice module. In the vertical, ECHAM6 consists

of 95 hybrid sigma-pressure levels up to the model top at 0.01 hPa and the Model for Ozone and Related Chemical Tracers (MOZART3; Kinnison et al. (2007)) is implemented in ECHAM6 (ECHAM6-HAMMOZ; Schultz et al. (2018)) to simulate





the chemical processes in the atmosphere. The horizontal resolution of the atmosphere is approximately 1.8° by 1.8° and the QBO can be internally generated. More details of FOCI can be found in Matthes et al. (2020).

The EMAC CCM integrates modules for tropospheric and middle atmospheric processes, including interactions with the ocean, land surfaces, and anthropogenic factors (Jöckel et al., 2010). Built on the Modular Earth Submodel System (MESSy2), EMAC links software codes from various institutes. Its core atmospheric model is the European Centre Hamburg General Circulation Model, 5th generation (ECHAM5) (Roeckner et al., 2006). For the SOLCHECK project, EMAC (ECHAM5 Version 5.3.02, MESSy Version 2.52) was operated in T42L47MA, extending to the mesopause (0.01 hPa). Key submodels for SOLCHECK include MECCA (Sander et al., 2011) for atmospheric chemistry, JVAL for photolysis rates (Sander et al., 2014), RAD/RAD-FUBRAD for radiation transfer (Dietmüller et al., 2016), QBO (Giorgetta and Bengtsson, 1999), UBCNOx for auroral influence (Funke et al., 2016) and MPIOM (Jungclaus et al., 2006) as an ocean component, operating at GR15L40 resolution.

The MPI-ESM-HR (Müller et al., 2018) consists of the atmosphere model European Centre Hamburg Model of version 6.3 (ECHAM6.3; Stevens et al. (2013)) at approximately 100 km horizontal resolution and with 95 levels in the vertical up to the top at 0.01 hPa (T128L95). It is coupled to the Max Planck Institute Ocean Model of version 1.6.3 (MPIOM; Jungclaus et al. (2013)) with a tripolar grid of about 0.4° and 40 vertical levels. Additional components of MPI-ESM-HR are the ocean bio-geo-chemical model Hamburg Model of the Ocean Carbon Cycle (HAMOCC; Ilyina et al. (2013)) and the land surface model Jena Scheme for Biosphere Atmosphere Coupling in Hamburg (JSBACH; Reick et al. (2013)). It is worth noticing that the MPI-ESM-HR is not a CCM and therefore uses prescribed ozone fields in the model.

## 2.2 Experimental design

Three sets of CMIP6 historical-like ensemble simulations are performed with the three climate models (i.e., FOCI, EMAC, and MPI-ESM-HR) driven by identical external forcing as recommended for CMIP6 (Eyring et al., 2016), except for the solar forcing. The first ensemble, named *FULL* in this study, includes full solar variability by using the CMIP6 solar forcing dataset (Matthes et al., 2017). The second ensemble only considers the low-frequency solar variability (hereafter *LOWFREQ*), in which the solar forcing was low-pass-filtered by a 33-year running mean. The last ensemble called *FIX* serves as a reference experiment, in which the solar forcing is fixed to 1850 preindustrial conditions. All ensemble simulations based on the three models have been integrated over the historical period 1850 – 2014 and the individual ensemble members have been initialized from different model years of a multi-centennial pre-industrial control simulation. 9 members of the *FULL* ensemble are performed with FOCI, 6 members with EMAC, and 10 members with MPI-ESM-HR. Thus, a total of 25 members and 4125 model years of the *FULL* ensemble have been analyzed in this study. The number of ensemble members of the *FIX* and the *LOWFREQ* ensembles are identical to the *FULL* ensemble, except for 8 members with MPI-ESM-HR. In this study, we mainly focus on assessing the 11-year solar signal in the middle atmosphere by using the *FULL* ensemble and comparing it to the *FIX* ensemble. The *LOWFREQ* ensemble is only used to compare our result with the work of Drews et al. (2022).

To validate our model results, 3D temperature and zonal wind from the ECMWF Reanalysis v5 (ERA5; Hersbach et al. (2020)) covering 1950 to the present are used. The December-January-February mean (DJF-mean) F10.7 radio flux in the solar





flux units (1 sfu = $10^{-22}$ Wm$^{-2}$ Hz$^{-1}$) from the CMIP6 solar forcing dataset (Matthes et al., 2017) is used as an index for solar variability.

## 2.3 Analysis methods

To investigate the stability of the solar imprints, we calculate correlation coefficients between the solar index F10.7 and several
meteorological variables (e.g., SWHR, temperature, ozone volume mixing ratio, and zonal wind) within a 45-year running window from 1850 to 2014 (121 windows in total). A standard deviation of the DJF-mean F10.7 index in the 45-year running window is used to indicate the mean strength of solar variability in that "time" window (called solar cycle amplitude hereafter). A scatter diagram of the correlation coefficients and the solar cycle amplitudes in all windows is used to demonstrate the possible reliance of the solar imprint on the strength of solar variability. Similarly, the averages of wind speed and temperature
in the 45-year running windows are scattered together with the correlation coefficients to show the possible dependence of the solar imprint on the middle atmosphere climate states. The 95% significance level for the correlation coefficient was calculated based on a two-tailed Student's t-test and the effective degrees of freedom in a 45-year window are calculated with the same method as used by Huo et al. (2023).

Composites of the meteorological variables based on the 11-year solar cycle are generated by calculating differences be-
tween the average values in all solar maximum years and minimum years. Here the solar maximum (minimum) years are defined following the method used in the work of Drews et al. (2022), i.e., the solar maximum (minimum) for each solar cycle includes three years — the year of the peak (valley) and two years around it. A 1000-fold bootstrapping test with replacement (Diaconis and Efron, 1983) is used to estimate the 90% statistical significance of the ensemble mean composites. The meridional temperature gradient is calculated by performing a centered finite difference operation on the latitude dimension.
Here we use annual mean data to examine the direct solar signal in the SWHR, temperature, and ozone volume mixing ratio anomalies. Then we focus on the NH extended winter season (from November to March) to investigate the possible resulting dynamic responses. For each meteorological variable from individual ensemble members and the observation, its least squares quadratic trend and the mean value of the whole data period are removed to get its detrended anomaly.

## 3 Results

### 3.1 Initial 11-year solar signals in the middle atmosphere

The incoming solar radiation absorbed by the Earth's middle and upper atmosphere has direct influences on the SWHR, temperature, and ozone production in the tropical region. Although solar UV radiation accounts for only a small part of the solar spectrum at the top of the atmosphere (approximately 8.73%) (Gueymard, 2004), it has a larger variation (of up to 6%) between the solar maximum and minimum of the 11-year solar cycle than the total solar irradiance (be about 0.07%) (Gray
et al., 2010). Figure 1 shows a scatter diagram of the annual mean SWHR (Fig. 1a) and temperature (Fig. 1b) anomalies at the tropical stratopause (around 1 hPa) in the *FULL* experiment vs. DJF-mean F10.7. When the F10.7 is smaller than 180 sfu,



both the SWHR and the temperature increase along with the enhancement of solar radiation in all models and most ensemble members, as expected. However, the linear increases of SWHR and temperature anomalies stop when the F10.7 is larger than 180 sfu. Please notice that the anomaly of the meteorological variable (i.e., SWHR and temperature here) is defined as its

deviation from the detrended mean value of the whole period. So the 0 value of the y-axis of Fig. 1 indicates that the value of that data point has no deviation from the mean value. More positive anomalies of the SWHR and temperature at approximate F10.7 $\geq$ 140 sfu suggest a possible solar impact when solar radiation is large. Most members from FOCI and EMAC show positive SWHR and temperature anomalies with fluctuating deviations at F10.7 $\geq$ 180 sfu. A very similar characteristic is also found in the ozone response at 10 hPa (as shown in Fig. 1c), suggesting a nonlinear response can take place in both the

tropical stratopause and the middle stratosphere when the solar forcing is strong enough (e.g., F10.7 $\geq$ 180 sfu in this study). The above responses of the SWHR, temperature, and ozone anomalies to the solar radiative forcing are absent in the *FIX* experiment (as shown in Fig. A1). However, an analogous fluctuation in the ensemble mean SWHR and temperature responses at the F10.7 $\geq$ 180 sfu, is found in both the *FULL* and *FIX* experiments (squares in Figs. 1a–b and Figs. A1a–b), implying a robust fingerprint of other external forcings in the tropical stratopause, in addition to the solar forcing. Significant responses

of the annual mean SWHR, temperature, and ozone anomalies in the tropical upper stratosphere to the 11-year solar cycle (i.e., composite difference between the solar maximum and minimum) can be simulated by all three models used in this study (as shown in Fig. A2), which is comparable with previous studies (Kunze et al., 2020; Matthes et al., 2017). However, FOCI (black line in Fig. A2) simulated larger SWHR and temperature responses around the tropical stratopause and a stronger ozone response in the ozone layer than EMAC (red line in Fig. A2). This could be due to different chemistry schemes (as described

in section 2.1) and shortwave radiation codes implemented in these two models. We should notice that the direct solar signal in the SWHR and ozone anomalies in MPI-ESM-HR are not shown here because the ozone response is prescribed and the output of SWHR is not available.

The different behavior of the SWHR, temperature, and ozone anomalies in the *FULL* and *FIX* experiments (Fig. 1 and Fig. A1) indicate significant solar fingerprints in the tropical upper stratosphere. To examine the solar influence in the middle

atmosphere at the decadal timescale (i.e., the signal of the 11-year solar cycle), we calculate the correlation coefficients of the F10.7 solar flux with SWHR, temperature, and ozone anomalies in a 45-year running window from 1850 to 2014. As shown in Fig. A3a, a significant and robust 11-year solar signal exists in the SWHR anomalies averaged over the tropical stratopause with a very small ensemble spread for all models. However, the resulting temperature anomalies in the tropical stratopause for all models show a slightly weaker positive correlation with the F10.7 in the 45-year windows during the period of 1850 – 1920

than the later period after 1920 (Fig. A3b). The larger ensemble spread in the earlier period (before 1920) indicates a strong disturbance of the internal variability on solar imprints in the tropical stratopause temperature. The 11-year solar signal in the tropical ozone anomalies at 10 hPa is significant and robust in all the 45-year running windows for both FOCI and EMAC (Fig. A3c). Figure 2a shows some reliance of the detected solar signal on the solar cycle amplitude by a scatter distribution of the correlation coefficients between the F10.7 solar flux and the SWHR anomalies vs. the standard deviation (SD) of the F10.7

index in all 45-year windows. A significant and robust 11-year solar signal in the SWHR anomalies (i.e., positive correlation coefficients) can be achieved in all members and all models in all 45-year windows (Fig. 2a). However the positive responses



in the temperature anomalies at 1 hPa (Fig. 2b) and ozone anomalies at 10 hPa (Fig. 2c) indicated by the positive correlation coefficients are only significant and robust when the SD is larger than 28. These positive responses in SWHR, temperature and ozone anomalies linearly increase along with the enhancement of the solar cycle amplitude when the solar cycle amplitude is smaller than 40 SD. This linear reliance of the detected solar signal on the solar cycle amplitude turns into non-linear features for all the models when the solar cycle amplitude is larger than 40 — more than one correlation coefficient can be achieved between the anomalous SWHR/temperature/ozone) and the F10.7 with an identical solar cycle amplitude. All the correlation coefficients are above the 95% significance level at SD $\geq$ 40 and the solar signal is more prominent in the ensemble mean.

We performed a similar analysis for the December temperature anomalies in the middle stratosphere (i.e., the air temperature at 10 hPa), which is partly related to solar-induced ozone anomalies. Although all three models simulated a positive temperature response in the ensemble mean (squares and solid lines in Fig. 3, none of them passes the 95% significance level in all the 45-year windows. The temperature anomalies at 10 hPa in FOCI only have robust positive correlations with the F10.7 index when the solar cycle amplitude is larger than 30 (Fig. 3a). However, the positive correlation of temperature anomalies at 10 hPa with the F10.7 in EMAC and MPI-ESM-HR show less reliance on the solar cycle amplitude in all the 45-year windows (Fig. 3b). Therefore, different from the direct solar signals in the tropical stratopause, the temperature response in the middle stratosphere (i.e., 10 hPa in this study) seems model-dependent and could be influenced by strong internal variability, like QBO and El Niño–Southern Oscillation.

### 3.2 The top-down mechanism in multiple-model ensemble simulations

According to previous work, the tropical upper stratospheric warm response to the solar cycle is expected to lead to an increased meridional temperature gradient and hence a so-called top-down dynamical response during the NH winter season (Kodera and Kuroda, 2002; Matthes et al., 2006; Drews et al., 2022; Kuroda et al., 2022). However, inconsistent results of the solar UV-forcing have been demonstrated in previous single-model studies, including various timings in the middle atmospheric response and an unstable solar-NAO connection. In this subsection, we re-examine the top-down mechanism with our multiple-model ensemble simulations.

Figure 4 shows composite differences between the solar maximum and minimum of the ensemble mean zonal-mean temperature anomalies and the anomalous poleward temperature gradients in the *FULL* experiment. All three models used in this study capture the significant warm response in the tropical and subtropical lower mesosphere and upper stratosphere (color shading contours in Fig. 4) during the extended winter season (i.e., from November to March). The solar-induced tropical stratopause warming is stronger in FOCI (first row of Fig. 4) than in EMAC (second row of Fig. 4), which is highly related to the SWHR response in the atmosphere layers above approximately 4 hPa (as shown in Fig. A4). Besides, a secondary lower-stratosphere warm response around 70 hPa can be found in both FOCI (first row of Fig. 4) and MPI-ESM-HR (third row of Fig. 4), but it disappears in EMAC (second row of Fig. 4). Previous work suggested the lower-stratosphere warm response is an indirect effect of the maximum solar forcing, involving changes in the ozone chemistry and the Brewer-Dobson circulation (Kodera and Kuroda, 2002; Gray et al., 2010; Mitchell et al., 2015), and nonlinear interference with internal variability (e.g. the QBO).





As the ozone responses with a maximum at 10 hPa are quite consistent in the three models (Fig. A5), here we think the nudged
       QBO in EMAC may mask the ensemble mean temperature response in the lower stratosphere (around 70 hPa) to some extent.

       Although the upper stratospheric warm response appears in the tropics and subtropics for all three models, the resulting
       changes in the meridional temperature gradient are quite different, as shown by the contours in Fig. 4. The anomalous poleward
       meridional temperature gradients increase in the lower mesosphere and upper stratosphere subtropics during all winter months
in FOCI (first row of Fig. 4), which are weaker in EMAC (second row of Fig. 4) due to its weaker responses of the tropical
       stratospheric SWHR and temperature anomalies. The meridional temperature gradient anomalies in the NH high latitude and
       polar region increase in FOCI during the November-December-January with a poleward and downward movement. A similar
       and weaker response of the temperature gradient anomaly also shows up in the *FULL* experiment with MPI-ESM-HR (third
       row of Fig. 4) but not in EMAC (second row of Fig. 4). The enhancement of the poleward meridional temperature gradient
anomaly leads to anomalous westerly winds in the NH high latitude and polar region during November-December-January in
       both FOCI and MPI-ESM-HR but not in EMAC (Fig. 5). It is worth noting that the enhanced meridional temperature gradient
       anomaly in the stratosphere in March (contours in the right column of Fig. 4) leads to a stronger stratospheric polar night jet for
       all three models (color shading contours in the right column of Fig. 5). The conformity of the meridional temperature gradient
       anomalies and zonal-mean zonal wind anomalies (Fig. 4 and Fig. 5), suggests a dominant role of the thermal-wind relationship
in the middle atmosphere. The downward movements of the zonal-mean meridional temperature gradient anomalies and the
       zonal-mean zonal wind anomalies in the polar vortex region from November to January could be a result of the interaction
       between the mean flow and upward planetary waves (Kodera and Kuroda, 2002; Matthes et al., 2006).

       Drews et al. (2022) proposed that the statistical top-down propagation of the 11-year solar cycle signal and its surface
       imprints can only be detected in an epoch with strong solar cycle amplitude (1932 – 2014) based on the CESM-WACCM.
However, this point could not be confirmed in the study of Spiegl et al. (2023) who analyzed a different set of simulations with
       the MPI-ESM-HR model w.r.t the MPI-ESM-HR contribution in this study. As the solar signals in the *FULL* experiment with
       FOCI also seem to be sensitive to the magnitude of solar cycle forcing (Fig. 2 and Fig. 3a), we repeated the above composite
       analysis for FOCI for the same weak and strong epochs as defined in the work of Drews et al. (2022). As shown in Figs. A6 and
       A7, both the zonal-mean ozone and temperature anomalies show a larger response in the strong epoch than in the weak epoch
for the whole extended winter season. The temperature responses lead to a different feature in the meridional temperature
       gradient anomalies in the weak and strong epochs, as shown by the contours in Fig. A7. The significant enhancements of the
       meridional temperature gradient anomalies are confined in the middle atmosphere (above the tropopause) and no clear top-
       down migration of the signals is found in the weak epoch (contours in the first row of Fig. A7). The top-down mechanism is
       confirmed in the strong epoch that both the positive meridional temperature gradient anomalies (contours in the second row of
Fig. A7) and the resulting zonal-mean zonal wind anomalies (second row of Fig. A8) are transferred from the stratosphere to
       the surface during the extended winter season (November–February). The different responses in the weak and strong epochs
       with FOCI are in line with the work of Drews et al. (2022) but cannot be found in the *FULL* experiment with EMAC and
       MPI-ESM-HR, implying a model-dependence of the simulated responses.



### 3.3 Model-dependent dynamical responses to the 11-year solar cycle forcing

As demonstrated by Fig. 5, responses in the stratospheric zonal-mean zonal wind anomalies show a large diversity between multiple models. To further explore possible reasons for the simulated "inconsistent" dynamical responses in the middle atmosphere, we use here the zonal-mean zonal wind anomalies averaged over 60°N – 65°N at 1 hPa and 10 hPa to approximately indicate the PNJ anomalies in the upper and middle stratosphere. Figure 6 shows the scatter plots of correlation coefficients between December PNJ anomalies and the F10.7 index vs. solar cycle amplitudes in all 45-year running windows. As a result of the significant warming response in the tropical stratopause (Figs. 2b), a strengthened PNJ (i.e., positive correlation coefficients) shows up in most of the ensemble members with the three models. We should also notice the manifestly negative correlation coefficients in FOCI and MPI-ESM-HR (Figs. 6a and c) but positive correlation in EMAC (Fig. 6b) when the solar cycle amplitude is very small (e.g., SD = 20). These "opposing" model results suggest a weak solar activity leading to a weak temperature response is no longer dominant in the correlation between the PNJ and the solar cycle and the effects of other internal variability become visible leading to a very large model spread. The positive correlation coefficients of the PNJ anomalies and the F10.7 also show a nonlinear reliance on the solar cycle amplitude at SD $\geq$ 40 for all three models, which is consistent with the significant and multiplex positive temperature response at the SD $\geq$ 40 (Fig. 2b). Although the tropical stratospheric ozone and temperature responses at 10 hPa are robust in FOCI and MPI-ESM-HR (Fig. 2c and Fig. 3), no significant (or robust) PNJ responses appear in the middle stratosphere (i.e., at 10 hPa here), as shown by Figs. 7a and c.

As supposed in Spiegl et al. (2023), the background states of the middle atmosphere may play a role in the initial solar signal transfer. To explore this point, we plotted the correlation coefficients between the December PNJ anomalies and the F10.7 index vs. the PNJ strengths (wind speed of zonal-mean zonal wind averaged over 60°N – 65°N), as shown in Fig. 8 and Fig. A9. Compared to the mean wind speed from the ERA5 (brown vertical line in Fig. 8, both EMAC and MPI-ESM-HR have a weaker PNJ (Figs. 8b and c), while it is stronger in FOCI (Fig. 8a). The positive and negative correlation coefficients from all ensemble members and all 45-year windows in FOCI and MPI-ESM-HR are distributed evenly regardless of the wind speed, showing very little sensitivity of PNJ responses to the PNJ strength in these two models. But when the 45-year mean wind speed is larger than 56 m/s in FOCI (Fig. 8a), a robust and weakened PNJ response (i.e., negative correlation coefficients in the ensemble mean and most of the ensemble members) appears, which is opposite to the proposed top-down mechanism. This "unexpected" dynamical response under a very strong PNJ background condition with FOCI implies that the model bias may lead to a false representation of the solar signal. The model-dependence of the PNJ response is also revealed in EMAC (Fig. 8b) where a robust and strengthened PNJ response appears when the simulated wind speed is close to the observed value (i.e. ERA5 in this study). As shown in Fig. A9, similar features of the simulated PNJ response are also found in the middle stratosphere (10 hPa), hinting at a thermally controlled effect in the upper and middle stratospheric dynamic response (i.e. the thermal-wind balance).

Here we calculated the zonal mean meridional temperature gradient ($\Delta T$) at 1 hPa and 10 hPa by using the mean value of the tropical box (25°S – 25°N) minus the mean value of the pole box (65°N – 90°N) for all three models (Table 1). The simulated meridional temperature gradients at 1 hPa in all three models (first row of Table 1) are weaker than the EAR5 ($\Delta T$





= 19.1 K), especially for the MPI-ESM-HR model where $\Delta T$ is just 7.8 K. The $\Delta T$ at 10 hPa in FOCI and EMAC (second
row of Table 1) are close to the value in EAR5 ($\Delta T = 23.6$ K) but it is still too weak in MPI-ESM-HR, implying a benefit from
the inclusion of the interactive chemistry in climate models (i.e. FOCI and EMAC here). The negative biases of the meridional
temperature gradients in EMAC and MPI-ESM-HR could be a reason for the weak PNJs in these models.

To further explore the possible causes of the model biases in the meridional temperature gradient and their influences on
the solar fingerprints in the PNJ anomalies, we show scatter plots of the correlation coefficients between the PNJ anomalies
at 1 hPa and the F10.7 index for all 45-year windows vs. temperature anomalies at 1 hPa averaged over the polar box (Fig. 9)
and tropical box (Fig. 10). As shown in Fig. 9, all three models have a warm bias in the pole region (65°N – 90°N) compared
to the ERA5, which is larger than 10 K in MPI-ESM-HR. This warm pole bias in general is the main reason for the weak
poleward meridional gradient ($\Delta T$) in all three models. Different from the polar region, the simulated temperature in the
tropical stratopause (i.e. the tropical box at 1 hPa) shows a slight positive bias in FOCI and a negative bias in EMAC (Figs.
10a and b). There is no bias in the tropical stratopause temperature in MPI-ESM-HR (Fig. 10c), and hence the warm pole bias
in this model is the main reason for its weak poleward meridional gradient ($\Delta T$). The negative bias of the tropical stratopause
in EMAC is partly responsible for the weak $\Delta T$ in this model. The positive correlation coefficients of the ensemble mean
December PNJ anomalies with the 11-year solar cycle are replaced by negative correlation coefficients in FOCI and MPI-
ESM-HR when the warm biases in the pole box (Figs. 9a and c) and the tropical box (Figs. 10a and c) reach their maxima.
In addition, the positive responses in the ensemble mean PNJ anomalies in all three models are not significant. However, the
positive response of the upper stratospheric PNJ anomalies (1 hPa) in EMAC increases and is more robust when the warm bias
in the tropical stratopause decreases (Fig. 10b).

Although a warm bias exists in all three models, the PNJ responses in the middle stratosphere (at 10 hPa shown here) are
not sensitive to the temperature of the pole region (Fig. 11). There is no robust PNJ response to the solar cycle forcing in FOCI
and MPI-ESM-HR as suggested by the approximately even distribution of the positive and negative correlation coefficients
(Figs. 11a and c), but a robust positive PNJ response can be found in EMAC (Fig. 11b). Figure 12 shows that FOCI has a warm
bias ($\geq 2$ K) in the tropical middle stratosphere (at 10 hPa). The positive PNJ response in the *Full* ensemble mean with FOCI
(i.e. positive correlation coefficients) decrease when its tropical warm bias increases (Fig. 12a) and the sign reverses when the
tropical warm bias is larger than approximately 4 K (Fig. 12a). The robust and positive PNJ response in EMAC is enhanced
when the tropical temperature increases (Fig. 12b). However, there is no robust PNJ response in the middle stratosphere with
MPI-ESM-HR and it is not sensitive to the simulated thermal state (Figs. 11c and 12c).

To summarize, a strengthened PNJ response to the 11-year solar cycle shows up in the upper stratosphere (1 hPa) in FOCI and
MPI-ESM-HR when the warming response in the tropical stratosphere is robust and significant. In EMAC, the upper and middle
stratospheric PNJ responses are not sensitive to the magnitude of the solar variability but are influenced by the background
states (e.g., wind speed of the mean flow). The cold bias in the tropical stratopause of EMAC likely dampens the initial solar
signal, and together with the warm bias in the polar region, they lead to a weak meridional temperature gradient and a weak
PNJ in this model. FOCI has a warm bias in both the tropics and the north pole region of the upper and middle stratosphere,
which may be responsible for the sensitivity of the initial solar signals to the magnitude of the solar forcing. However, the





too strong PNJ in FOCI possibly dampens the upward propagation of planetary waves and hence influences the downward
propagation of the initial solar signal in the winter season.

## 4 Conclusions

This study aimed to assess the 11-year solar cycle signal in the middle atmosphere since the detected solar imprints in previous studies still show large discrepancies among different climate models and the solar signal in the relativley short period with reliable observations is hard to distinguish from other signals or internal variability. For this purpose, three sets of CMIP6
historical-like large ensemble simulations — the *FULL*, *FIX*, and *LOWFREQ* — were performed with two CCMs (i.e., FOCI and EMAC) and one high-top climate model without interactive chemistry (i.e., MPI-ESM-HR). Each set of simulations is forced by identical CMIP6 external forcings and a different solar forcing — full solar variability in the *FULL*, no solar variability in the *FIX*, and only low-frequency of the solar variability in the *LOWFREQ*. Comparisons between the *FULL* and the *FIX* ensembles allow us to derive the possible impacts of solar forcing, and the ensemble mean of each set can extract the
solar signals to some extent.

Our results show a robust and significant solar imprint on the SWHR and temperature anomalies in the upper stratosphere, as well as in the ozone anomalies in the middle stratosphere (in the two CCMs). FOCI simulated stronger 11-year solar cycle signals in the SWHR and temperature anomalies in the upper stratosphere above 4 hPa and a larger ozone response than EMAC. When the solar cycle amplitude, indicated by the standard deviation of the DJF-mean F10.7 in the 45-year running window,
is smaller than 40, the responses of the SWHR, temperature, and ozone anomalies in the tropical upper stratosphere with all three models show linear reliance on the solar cycle amplitude (i.e., their positive correlation coefficients increase along with the solar cycle amplitude). However, the reliance of the detected solar signals in the SWHR, temperature, and ozone anomalies on the strength of solar activity is more complex at SC $\geq 40$. The middle stratospheric temperature response to the 11-year solar cycle simulated by FOCI is sensitive to the solar cycle amplitude, which is not the case for EMAC and MPI-ESM-HR.
Although all three models simulated a warm response in the tropical upper stratosphere to the 11-year solar cycle forcing, the responses in the poleward meridional temperature gradient as well as the zonal-mean zonal wind anomalies are quite different between the models. The top-down mechanism that has been claimed to explain the downward propagation of the initial solar signals transport from the middle atmosphere to the troposphere can be found in the ensemble mean of the *FULL* with FOCI, and the responses are more significant in a strong epoch with large solar cycle amplitude. However, the top-down response
in the ensemble mean of *FULL* is much weaker in MPI-ESM-HR and has no clear downward propagation in EMAC. These diverse results from multiple climate models suggest the 11-year solar cycle signal and its transport in the atmosphere are more complex than expected. The linear methods used in this study (i.e., ensemble mean and composite) may not be able to extract the 11-year solar cycle signal from the background noise (e.g., large internal variability in the zonal mean zonal wind in the PNJ region).
As discussed in the work of Kunze et al. (2020), the model dynamics may modulate the solar responses. Our further analysis with a particular focus on the December PNJ dynamical response suggests that model biases can influence the fingerprints of the 11-year solar cycle. A strengthened PNJ response (but not significant) to the solar cycle in the upper stratosphere (1 hPa)

can be found in all three models when the warming response in the tropical stratopause is robust and significant. No robust December PNJ response can be found in the middle stratosphere (10 hPa) in FOCI and MPI-ESM-HR. However, the simulated

PNJ response in EMAC is not sensitive to the solar cycle amplitude but is influenced by the background wind speed. The warm bias in the pole region and cold bias in the tropical stratopause of EMAC lead to a large negative bias in the meridional temperature gradient, which dampens the initial 11-year solar cycle signal in the tropical upper stratosphere and thereby limits the zonal-mean zonal wind responses in the high latitude and pole region. The large warm pole bias in the middle and upper stratosphere of the MPI-ESM-HR leads to a weak PNJ and may be responsible for the weak dynamical responses in this

model. In addition, the extreme tropical warm bias in the middle stratosphere of FOCI can result in a too strong zonal wind and a reversed (weakened) PNJ response.

*Code and data availability.* The simulation data produced for this study (i.e. outputs of the experiments listed in Table 1) are publicly available on DOKU at DKRZ (Wahl et al., 2023). The MPI-ESM-HR historical "FULL" simulations are available from the CMIP6 archive of the Earth System Grid Federation (https://esgf-data.dkrz.de/projects/cmip6-dkrz/) and the MPI-ESM-HR historical simulations with fixed

solar forcing ("FIX") from the World Data Center for Climate at DKRZ (Pohlmann, 2021). The ERA5 dataset (Hersbach et al., 2020) produced by the Copernicus Climate Change Service (C3S) at ECMWF, is available on the Climate Data Store (CDS). The solar index and all solar forcing data used in this study (Matthes et al., 2017) can be downloaded from https://solarisheppa.geomar.de/cmip6. Codes to reproduce the analysis and figures are archived from the corresponding author.

*Author contributions.* WH did the analysis and wrote the paper with input from all co-authors. TS performed the model simulations with

EMAC. SW performed the FOCI simulation. HP and JK performed the simulations based on MPI-ESM-HR. KM and UL initiated the study and assisted with the interpretation of the results. All authors commented on the manuscript

*Competing interests.* The corresponding author has declared that none of the authors has any competing interests.

*Acknowledgements.* We would like to thank all the scientists, software engineers, and administrators who contributed to the development

of the climate models used in this study (i.e. FOCI, EMAC, and MPI-ESM-HR). We are grateful for the computing support and resources provided by the Deutsche Klimarechenzentrum (DKRZ) in Hamburg, Germany, as well as the computing time granted by the Resource Allocation Board and provided on the supercomputer Lise and Emmy at NHR@ZIB and NHR@Göttingen as part of the NHR infrastructure.



This research has been funded by the Federal Ministry of Education and Research in Germany (ROMIC II-SOLCHECK project (grant no. 01LG1906A, 01LG1906B, 01LG1906C)).



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





**Table 1.** Meridional temperature gradients $\Delta T$ (Units: K) in climate models and the EAR5 for a common period of 1950 – 2014.

| December $\Delta T$ | FOCI | EMAC | MPI-ESM-HR | EAR5 |
|---|---|---|---|---|
| $\Delta T$ at 1 hPa | 14.5 | 11.4 | 7.8 | 19.1 |
| $\Delta T$ at 10 hPa | 24.4 | 21.6 | 15.4 | 23.6 |

$\Delta T$ is calculated by the mean value of the zonal mean temperature in the tropics ($25^\circ$S

-– $25^\circ$N) minus the mean value in the pole zone ($65^\circ$N -– $90^\circ$N).



**Figure 1. Direct solar signatures in the upper stratosphere.** (a) Scatter diagram of the annual-mean shortwave heating rate anomalies (units: Kday$^{-1}$) at the tropical stratopause (averaged over 25°S – 25°N at 1 hPa) vs. the F10.7 index from the *FULL* experiment with FOCI (black) and EMAC (red). Dots indicate individual members and squares represent the ensemble mean for each model. (b) as (a), but for temperature (K) from FOCI (black), EMAC (red), and MPI-ESM-HR (green). (c) as (a) but for the ozone volume mixing ratio (ppmv) at 10 hPa from FOCI (black) and EMAC (red).





**Figure 2. Initial 11-year solar cycle signals in the upper and middle stratosphere vs. the solar cycle amplitude.** (a) Scatterplot of correlation coefficients between the annual shortwave heating rate anomalies in the tropical stratopause (averaged over $25°S – 25°N$ at 1 hPa) and the F10.7 index in all 45-year running windows vs. the solar cycle amplitude (standard deviations of the F10.7 index in all 45-year windows) for FOCI (black) and EMAC (red) in the *FULL* experiment. Dots indicate the correlation coefficients of individual members and squares (marker lines) represent the ensemble mean for each model. The shadow regions indicate the spread of correlation coefficients among individual members (between maximum and minimum). The black dashed line indicates the 95% significance level. (b) is the same as (a), but for the temperature anomalies from FOCI (black), EMAC (red), and MPI-ESM-HR (green). (c) is the same as (a), but for the O3 volume mixing ratio anomalies at 10 hPa from FOCI (black) and EMAC (red).







**Figure 3. 11-year solar cycle signals in the middle stratospheric temperature anomalies vs. the solar cycle amplitude.** Scatter plots of correlation coefficients between the temperature anomalies in December averaged over 25°S – 25°N at 10 hPa and the F10.7 index in all 45-year running windows vs. the solar cycle amplitude for (a) FOCI (black), (b) EMAC (red), and (c) MPI-ESM-HR (green) in the *FULL* experiment. Dots indicate the correlation coefficients of individual members and squares represent the ensemble mean for each model. The shadow regions indicate the spread of correlation coefficients among individual members (between maximum and minimum). The black dashed line indicates the 95% significance level.





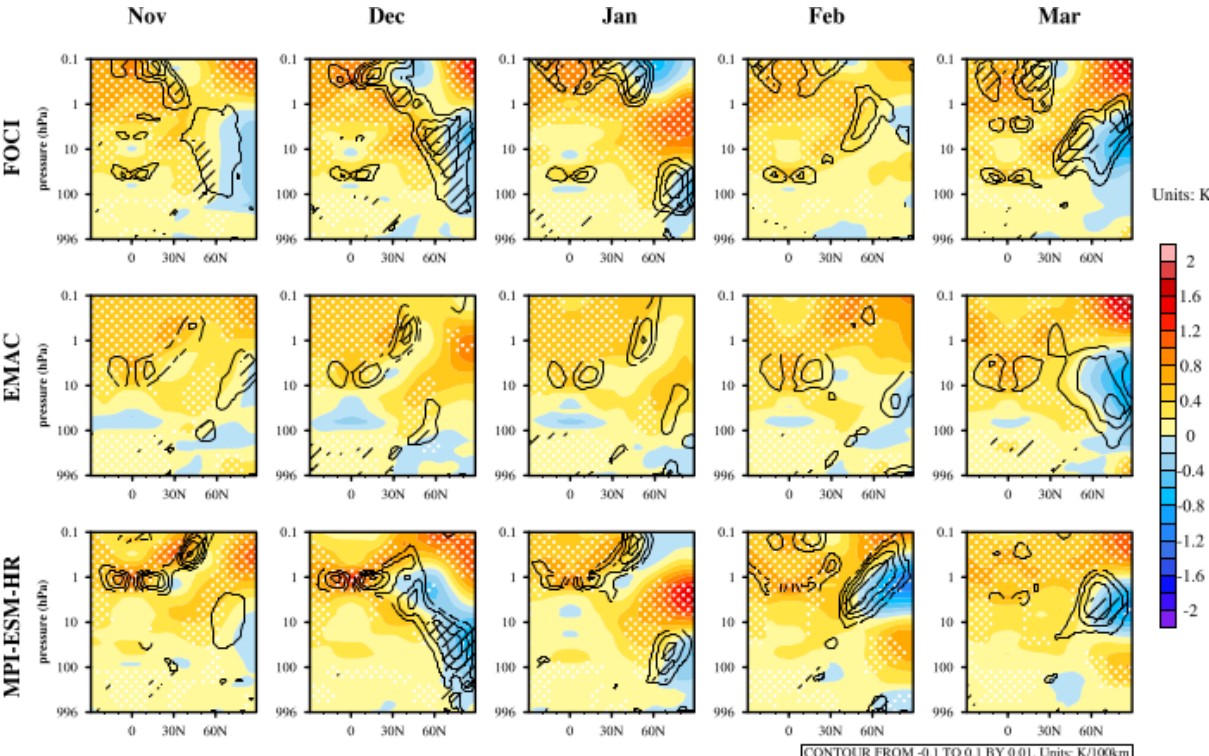

**Figure 4. Zonal-mean temperature anomalies response to the 11-year solar cycle forcing during the extended winter season (from November to March).** Composite differences between solar maximum and minimum of the ensemble mean zonal-mean temperature anomalies (Units: K, color shading contours) and the poleward meridional temperature gradients (Units: $K(100km)^{-1}$, contours) from the *FULL* experiment with FOCI (top panels), EMAC (middle panels), and MPI-ESM-HR (bottom panels). Latitude-height cross sections are from 30°S to 90°N and 996 hPa to 0.1 hPa. Only the positive meridional temperature gradient anomalies (poleward) are shown here. The 90% significance level for the composite of temperature (meridional gradients) anomalies is indicated by white dots (black hatching) based on a 1000-fold bootstrapping test.

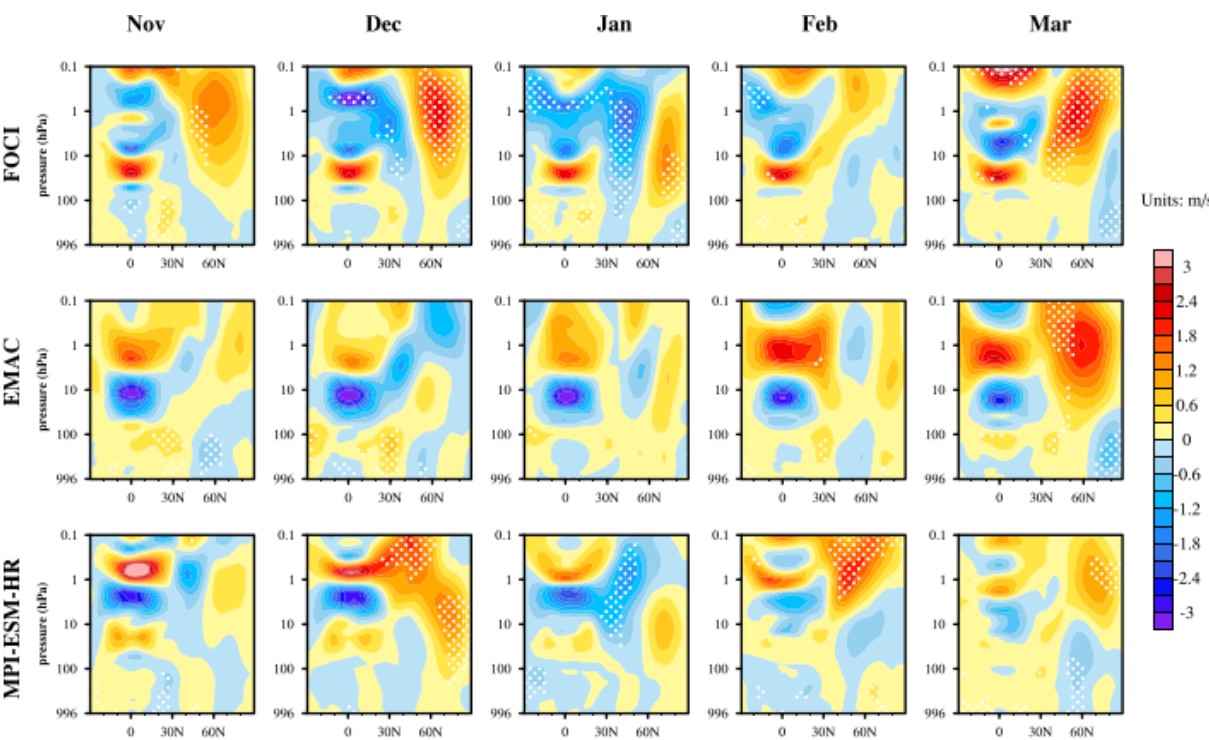

**Figure 5. Zonal-mean zonal wind anomalies response to the 11-year solar cycle forcing during the extended winter season (from November to March).** The same as Figure 4, but for the ensemble mean zonal-mean zonal wind anomalies (units: ms$^{-1}$) in the *FULL* experiment for FOCI (top), EMAC (middle), and MPI-ESM-HR (bottom).







**Figure 6. Upper stratospheric polar night jet anomalies response to the solar forcing vs. the solar cycle amplitude** Scatter plots of correlation coefficients between the December zonal-mean zonal wind anomalies averaged over $60°N – 65°N$ at 1 hPa and the F10.7 index in all 45-year running windows vs. the solar cycle amplitude for (a) FOCI (black), (b) EMAC (red), and (c) MPI-ESM-HR (green). Dots indicate the correlation coefficients of individual members and squares represent the ensemble mean for each model. The shadow regions indicate the spread of correlation coefficients among individual members (between maximum and minimum). The black dashed line indicates the 95% significance level.



**Figure 7. Middle stratospheric polar night jet anomalies responses to the solar forcing vs. the solar cycle amplitude.** Same as Figure 6, but for zonal-mean zonal wind anomalies averaged over $60°N – 65°N$ at 10 hPa.





**Figure 8. The upper stratospheric polar night jet responses to the solar forcing vs. the mean flow strength.** Scatter plots of correlation coefficients between the December zonal-mean zonal wind anomalies averaged over $60°N – 65°N$ at 1 hPa and the F10.7 index vs. the wind speed averaged over $60°N – 65°N$ at 1 hPa in all 45-year windows for (a) FOCI (black), (b) EMAC (red), and (c) MPI-ESM-HR (green). Dots indicate the correlation coefficients of individual members and squares represent the ensemble mean for each model. The shadow regions indicate the spread of correlation coefficients (between maximum and minimum) among individual members. The black dashed line indicates the 95% significance level. The brown vertical lines indicate $1950 – 2014$ mean wind speed from ERA5 averaged over $60°N – 65°N$ at 1 hPa.







**Figure 9. The upper stratospheric polar night jet responses to solar forcing vs. the polar stratopause temperature.** Scatter plots of correlation coefficients between the December zonal-mean zonal wind anomalies averaged over $60°N – 65°N$ at 1 hPa and the F10.7 index vs. Pole temperature averaged over $65°N – 90°N$ at 1 hPa in all the 45-year windows for (a) FOCI (black), (b) EMAC (red), and (c) MPI-ESM-HR (green). Dots indicate the correlation coefficients of individual members and squares represent the ensemble mean for each model. The shadow regions indicate the spread of correlation coefficients among individual members (between maximum and minimum). The black dashed line indicates the 95% significance level. The brown vertical lines indicate $1950 – 2014$ mean temperature from ERA5 averaged over the North Pole ($65°N – 90°N$) at 1 hPa.





**Figure 10. The upper stratospheric polar night jet responses to solar forcing vs. the tropical stratopause temperature.** Scatter plots of correlation coefficients between the December zonal-mean zonal wind anomalies averaged over $60°N - 65°N$ at 1 hPa and the F10.7 index vs. tropical stratopause temperature averaged over $25°S - 25°N$ at 1 hPa in all 45-year windows for (a) FOCI (black), (b) EMAC (red), and (c) MPI-ESM-HR (green). Dots indicate the correlation coefficients of individual members and squares represent the ensemble mean for each model. The shadow regions indicate the spread of correlation coefficients among individual members (between maximum and minimum). The black dashed line indicates the 95% significance level. The brown vertical lines indicate 1950 – 2014 mean tropical stratopause temperature from ERA5 averaged over $25°S - 25°N$ at 1 hPa.





**Figure 11. The middle stratospheric polar night jet responses to the solar forcing vs. the polar temperature.** Scatter plots of correlation coefficients between the December zonal-mean zonal wind anomalies averaged over 60°N – 65°N at 10 hPa and the F10.7 index vs. Pole temperature averaged over 65°N – 90°N at 10 hPa in all 45-year windows for (a) FOCI (black), (b) EMAC (red), and (c) MPI-ESM-HR (green). Dots indicate the correlation coefficients of individual members and squares represent the ensemble mean for each model. The shadow regions indicate the spread of correlation coefficients among individual members (between maximum and minimum). The black dashed line indicates the 95% significance level. The brown vertical lines indicate 1950 – 2014 mean Pole temperature from ERA5 averaged over 65°N – 90°N at 10 hPa.





**Figure 12. The middle stratospheric polar night jet responses to the solar forcing vs. the tropical temperature.** Scatter plots of correlation coefficients between the December zonal-mean zonal wind anomalies averaged over $60°N – 65°N$ at 10 hPa and the F10.7 index vs. tropical stratopause temperature averaged over $25°S – 25°N$ at 10 hPa in all 45-year windows for (a) FOCI (black), (b) EMAC (red), and (c) MPI-ESM-HR (green). Dots indicate the correlation coefficients of individual members and squares represent the ensemble mean for each model. The shadow regions indicate the spread of correlation coefficients among individual members (between maximum and minimum). The black dashed line indicates the 95% significance level. The brown vertical lines indicate 1950 – 2014 mean tropical stratopause temperature from ERA5 averaged over $25°S – 25°N$ at 10 hPa.



**Appendix A: Supplementary figures A1-A9 and table A1**







**Figure A1.** The same as Figure 1, but for the solar-fixed historical simulations (i.e. *FIX*.)





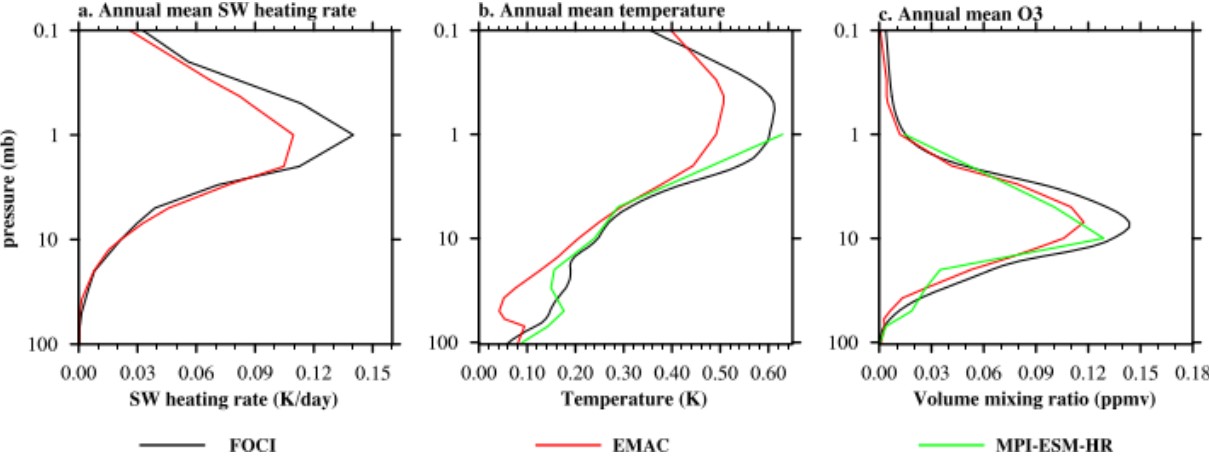

**Figure A2.** Composite differences between solar maxima and minima of the annual tropical (averaged over 25°S – 25°N) (a) short wave heating rate anomalies, (b) temperature anomalies, and (c) O3 volume mixing ratio anomalies in the *FULL* ensemble mean with respect to the *FIX* ensemble mean. Here black lines indicate results of FOCI, red lines are EMAC and green lines are MPI-ESM-HR.







**Figure A3.** (a) Correlation coefficients between the annual shortwave heating rate in the tropical stratopause (averaged over 25°S–25°N at 1 hPa) and the F10.7 index in a 45-year running window for FOCI (black) and EMAC (red). solid lines represent the correlation coefficients of the ensemble mean for each model and the shadow regions indicate the spread of correlation coefficients among individual members (between maximum and minimum). The black dashed line indicates the 95% significance level. (b) is the same as (a), but for the air temperature from FOCI (black), EMAC (red), and MPI-ESM-HR (green). (c) is the same as (a) but for the O3 volume mixing ratio at 10 hPa from FOCI (black) and EMAC (red).





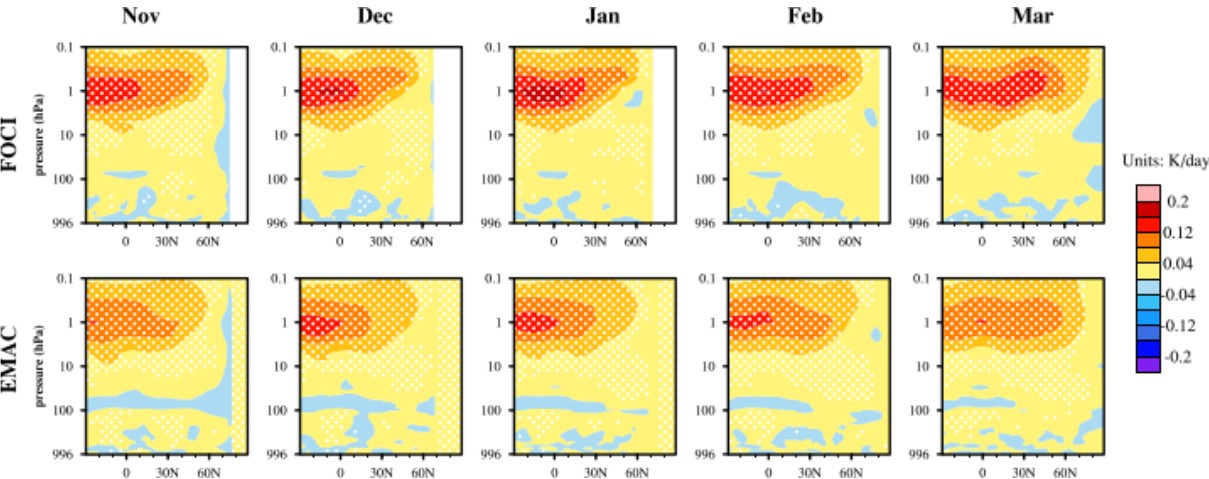

**Figure A4.** Composite differences between solar maxima and minima of the ensemble mean zonal-mean shortwave heating rate anomalies (Units: Kday$^{-1}$; color shading contours) in the *FULL* experiment with FOCI (top panels) and EMAC (bottom panels). Latitude-height cross sections are from 30°S to 90°N and 996 hPa to 0.1 hPa. The 90% significance level for the composite of air temperature anomalies is indicated by white dots based on a 1000-fold bootstrapping test.



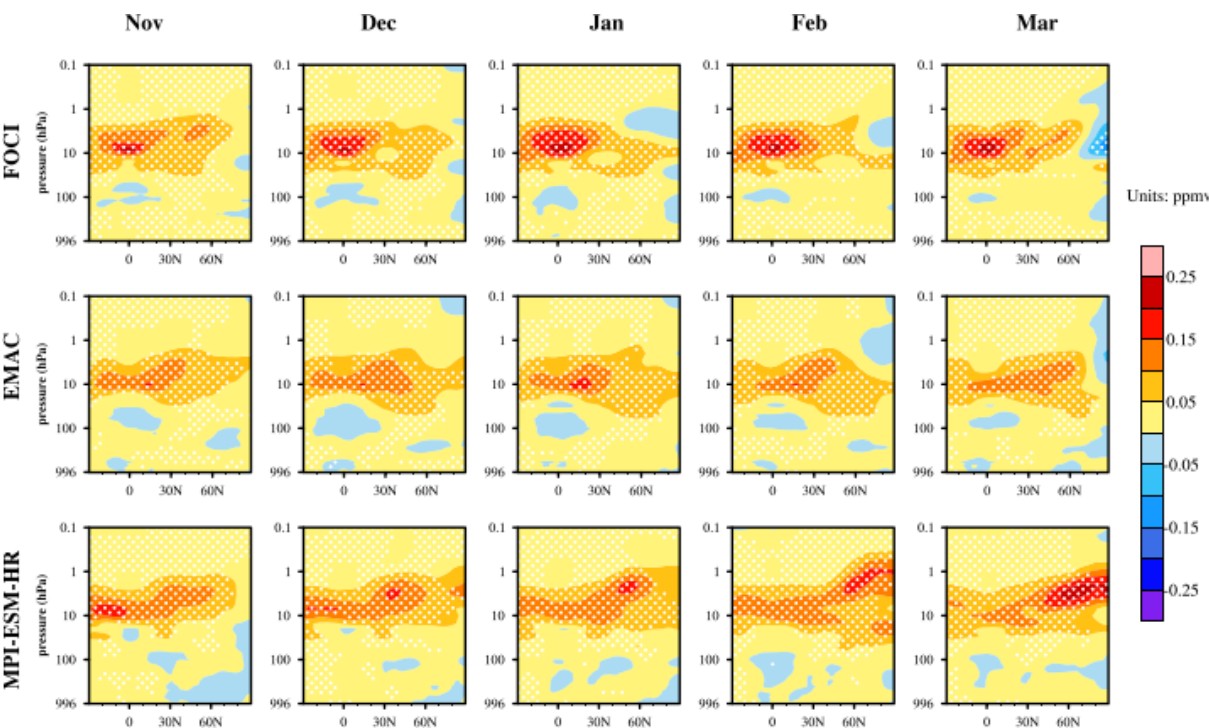

**Figure A5.** Composite differences between solar maxima and minima of the ensemble mean zonal-mean O3 volume mixing ratio anomalies (Units: ppmv; color shading contours) in the *FULL* experiment with FOCI (top panels), EMAC (middle panels), and MPI-ESM-HR (bottom panels). Latitude-height cross sections are from 30°S to 90°N and 996 hPa to 0.1 hPa. The 90% significance level for the composite of O3 volume mixing ratio anomalies is indicated by white dots based on a 1000-fold bootstrapping test.



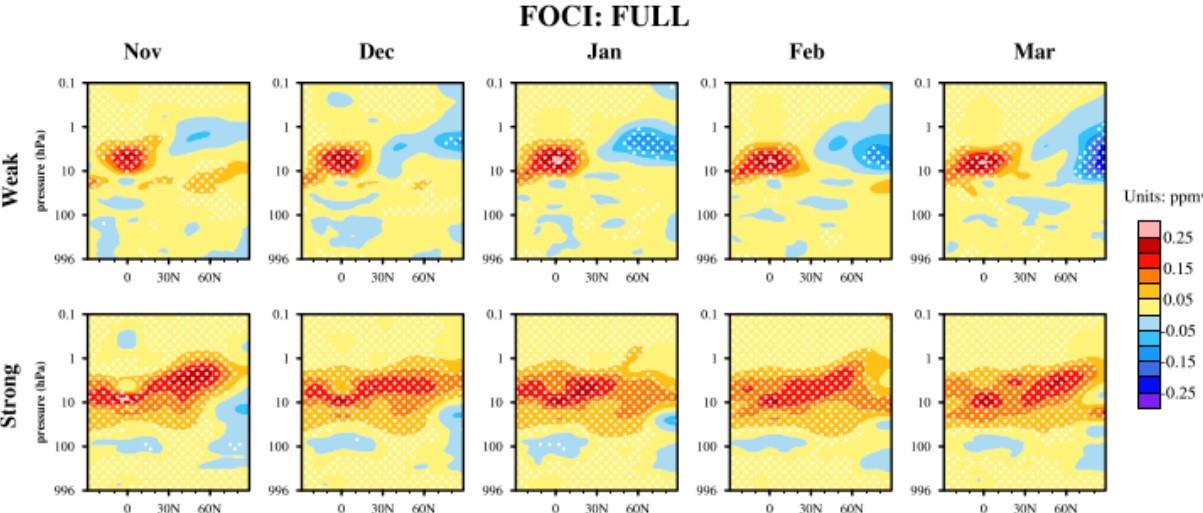

**Figure A6.** Composite differences between solar maxima and minima of the ensemble mean zonal-mean O3 volume mixing ratio anomalies (Units: ppmv; color shading contours) in the *FULL* experiment with FOCI during the weak (top panels) and strong solar epochs (bottom panels). Latitude-height cross sections are from 30°S to 90°N and 996 hPa to 0.1 hPa. The 90% significance level for the composite of air temperature anomalies is indicated by white dots based on a 1000-fold bootstrapping test.



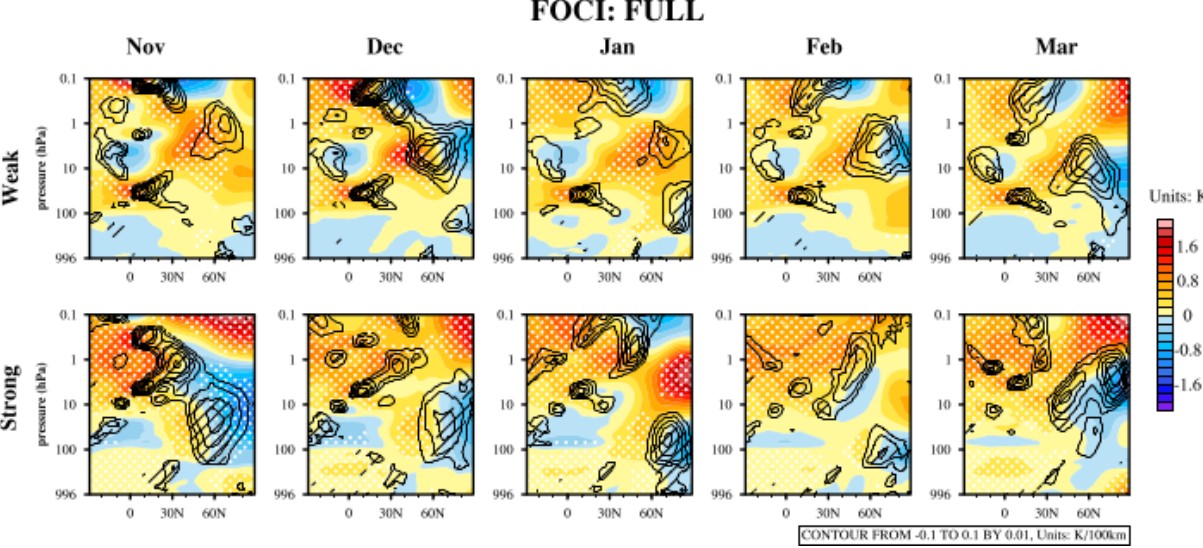

**Figure A7.** Composite differences between solar maxima and minima of the ensemble mean zonal-mean air temperature anomalies (units: K; color shading contours) and the poleward meridional temperature gradients (Units: K(100km)$^{-1}$; contours) in the *FULL* experiment with FOCI during the weak (top panels) and strong solar epochs (bottom panels). Latitude-height cross sections are from 30°S to 90°N and 996 hPa to 0.1 hPa. Only the positive meridional temperature gradient anomalies (poleward) are shown here. The 90% significance level for the composite of air temperature (meridional gradients) anomalies is indicated by white dots (black hatching) based on a 1000-fold bootstrapping test.





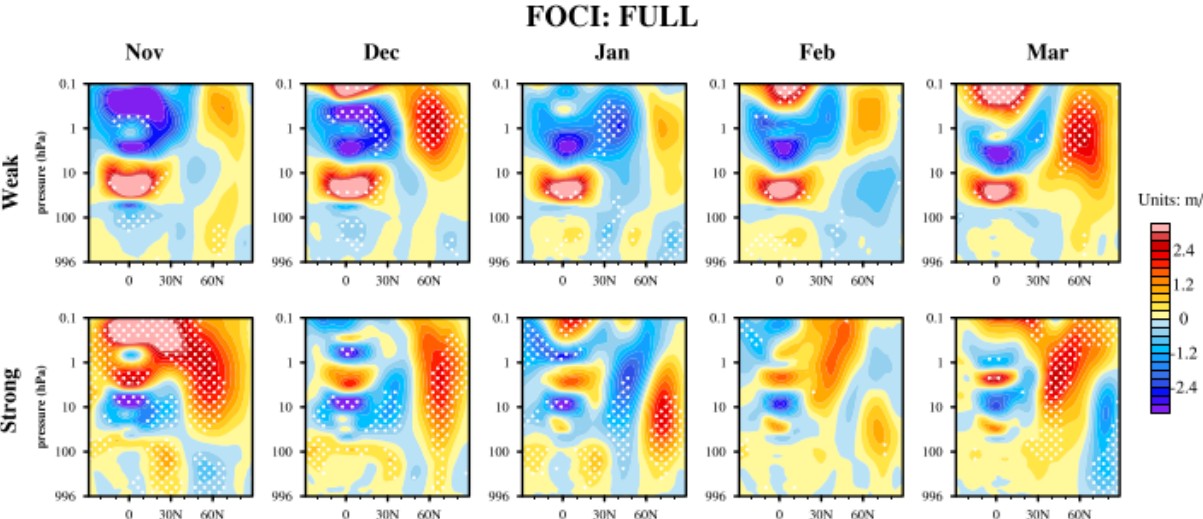

**Figure A8.** Composite differences between solar maxima and minima of the ensemble mean zonal-mean zonal wind anomalies (units: m/s; color shading contours) in the *FULL* experiment with FOCI during the weak (top panels) and strong solar epochs (bottom panels). Latitude-height cross sections are from 30°S to 90°N and 996 hPa to 0.1 hPa. The 90% significance level for the composite of zonal-mean zonal wind anomalies is indicated by white dots (black hatching) based on a 1000-fold bootstrapping test.





**Figure A9.** Composite differences between solar maxima and minima of the ensemble mean zonal-mean zonal wind anomalies (Units: m/s, color shading contours) in the *FULL* experiment with FOCI during the weak (top panels) and strong solar epochs (bottom panels). Latitude-height cross sections are from 30°S to 90°N and 996 hPa to 0.1 hPa. The 90% significance level for the composite of air temperature anomalies is indicated by white dots based on a 1000-fold bootstrapping test.