# Peer review of "Assessment of the 11-year solar cycle signals in the middle atmosphere during boreal winter with"

_EGUsphere, 2024_

## Author Comment (AC1)

General comments

This study assesses the 11-year solar cycle signals in the middle atmosphere in multiple-model ensemble simulations. This study starts with initial solar cycle signals in short wave heating rate, ozone, and temperature anomalies and continues with an analysis of whether the top-down mechanism explaining the downward propagation of these initial solar signals can be found in the presented models. I find the study highly relevant for long-term discussion of indirect solar effects and recommend it for publication with minor comments listed below.

Specific comments

I am not convinced that three sets of historical-like simulations with 9, 6 and 10 ensemble members, respectively, can be called "large ensembles" (l2+l330) since we do not know how large the large ensemble needs to be (Milinski et al, 2020).

→ Thanks for the reference. Indeed, as shown in the work of Drews et al. (2022), even a 10-member ensemble is still not large enough to quantify the solar signal in the zonal wind at stratopause (~1 hPa) or the surface. But it's good enough to quantify the solar signal in the temperature at the tropical stratopause (see Figure 14 in the Extended Data of Drews et al. (2022)). We deleted the word "large" to avoid misleading.

**Reference:**

Drews, A., Huo, W., Matthes, K., Kodera, K., and Kruschke, T.: The Sun's role in decadal climate predictability in the North Atlantic, Atmos. Chem. Phys., 22, 7893–7904, https://doi.org/10.5194/acp-22-7893-2022, 2022.

Can you specify the threshold in the abstract (l8)?

→ Revised. **Please see lines 7-8**.

What does "partly confirmed" (l25) mean?

→ It means that top-down propagation of the solar signal was found in subsequent studies, but with varying times of propagation. The texts are revised**, please see lines 26-29.**

The authors should elaborate more on the fact that the solar signal may not be stationary (Thejll et al, 2003) related to modulation by QBO and PDO (l30).

→ Thanks for the suggestion, the descriptions are revised. **Please see lines 29-33.**

I would omit the "controversial" label (l39) even though these studies may reduce the confidence level of the solar-NAO connection as you state.

→ Replaced the "controversial" with "diverse".

As shown in previous studies (e.g. Mitchell et al, 2014; Kuchar et al, 2015) the upper stratospheric equatorial temperature anomaly related to the solar cycle has been detected showed a statistically significant signal with structure and amplitude of 1–1.25 K. Temperature response in Fig. 2A maximizes at 0.6 K. I would say that models a bit underestimate the response even with comparison (l167) with Kunze et al (2020). These facts should be discussed and even analyzed more thoroughly in your models.

➔ Thanks for pointing this out. In general, I would say, yes, the climate models with interactive chemistry used in this study (i.e., FOCI and EMAC) might a bit underestimate the response compared to the model with prescribed ozone chemistry (i.e., MPI-ESM-HR). This is consistent with the results based on CMIP5 simulations (as shown in Figure 4 in the work of Mitchell et al (2015)) that the models with interactive ozone chemistry simulate a response of 0.5 K in the tropical stratopause.

Besides, we calculated the composite differences between solar maxima and minima of the annual tropical SWHR, temperature, and O3 volume mixing ratio anomalies in the FULL ensemble mean with respect to the FIX ensemble mean (Figure. 2A). Following the method used in the work of Drews et al. (2022), here three years — the year of the peak (valley) of each solar cycle and two years around it — are selected as solar maximum (minimum). This definition could avoid a problem from double "peaks" in one solar cycle and smooth out the high-frequency interannual variability (< 3 years). But it also might lead to an "underestimation" compared to other methods, like multiple linear regression used in the works of Mitchell et al (2015) and Spiegl et al. (2023). To check the influence of the method, we repeated the composite analysis only based on the peak and valley years (Table R1), and the results are shown in Figure. R1.

[Figure]

**Figure. R1.** Same as Figure. 2A, but composite based on the solar maximum and minimum years listed in Table R1 (below).

**Table R1.** The peak (maximum) and valley (minimum) years of solar cycles used for the "test" composite shown in Fig. R1

| Solar peak | 11, 21, 34, 44, 57, 68, 78, 88, 98, 108, 119, 131, 140, 151 |
| Solar valley | 6, 17, 28, 39, 52, 63, 73, 83, 94, 104, 114, 126, 136, 146 |

Comparison between Figure. R1 and Figure. A2, we found our method has very little impact on the responses in chemistry-climate models (FOCI and EMAC), but reduces the simulated temperature response in the MPI-ESM-HR model a lot (from 1.0 K to 0.6 K at the stratopause). Well, even the 1 K response of the tropical stratopause (Figure. R1) in the MPI-ESM-HR is still smaller than the result in the work of Spiegl et al. (2023) (about 1.2 K), which is estimated by a multiple linear regression method and CMIP5 historical simulations.

Compared to the reanalysis datasets (e.g., ERA-I, MERRA, and JRA-55 used in Mitchell et al (2015)), there is an "underestimation" of the initial solar signals in the upper and middle stratosphere in both the CMIP5 models (Figure 4 in Mitchell et al (2015)) and the models used in our study, especially for the models with interactive ozone chemistry. However, we should note that the reanalysis datasets do not cover the "weak" solar cycles (i.e., solar cycles before 1940) and only one member for each dataset. The SWHR is sensitive to the strength of the solar cycle (as demonstrated in Figure 1 of the work of Spiegl et al. (2023)) and hence a weaker solar signal is achieved when more "weak" solar cycles are included (e.g., back to 1850). In addition, the ensemble mean of transition simulations could smooth out some "coincide" between the solar signal and internal variability. In the study of Kunze et al (2020), the response of SWHR in tropical stratopause is about 0.2 $Kd^{-1}$ based on sensitivity simulations forced by perpetual solar maximum conditions of the solar cycle 22 maximum and 0.15 $Kd^{-1}$ in our transient simulations. Due to the difference in methods and experiments' design, we will not directly conclude that our models underestimate the solar signals but include a discussion on it. **Please see lines 167-182.**

**References:**

Mitchell, D., Misios, S., Gray, L., Tourpali, K., Matthes, K., Hood, L., Schmidt, H., Chiodo, G., Thiéblemont, R., Rozanov, E., Shindell, D., and Krivolutsky, A.: Solar signals in CMIP-5 simulations: the stratospheric pathway, Q.J.R. Meteorol. Soc., 141, 2390–2403, https://doi.org/https://doi.org/10.1002/qj.2530, 2015.

Spiegl, T. C., Langematz, U., Pohlmann, H., and Kröger, J.: A critical evaluation of decadal solar cycle imprints in the MiKlip historical ensemble simulations, Weather and Climate Dynamics, 4, 789–807, https://doi.org/https://doi.org/10.5194/wcd-4-789-2023, 2023.

Kunze, M., Kruschke, T., Langematz, U., Sinnhuber, M., Reddmann, T., and Matthes, K.: Quantifying uncertainties of climate signals in chemistry climate models related to the 11-year solar cycle – Part 1: Annual mean response in heating rates, temperature, and ozone, Atmos. Chem. Phys., 20, 6991–7019, https://doi.org/https://doi.org/10.5194/acp-20-6991-2020, 2020.

Based on Fig. 1.c (l159), the authors suggest that a nonlinear response can occur when the solar forcing is strong enough but I would soften these statements given the large spread and not enough samples for high sfu values.

➔ Thanks for the suggestion. The statements are revised. **Please see lines 162-166.**

I would omit the publications of Gray et al (2010) which provides a review of the Kodera and Kuroda mechanism and Mitchell et al (2015; CMIP5) which does not show any BDC response (l219) and only highlight the link between weaker BDC and lower-stratospheric temperature induced by the 11-year solar cycle.

➔ Revised.

How different (l240)?

➔ Spiegl et al. (2023) analyzed a set of historical simulations forced by CMIP5 external forcings (i.e., the CMIP5 protocol) and the solar forcing follows the reconstruction of lean (2000). The simulations were integrated from 1850 to 2005 and they focused on the period 1880 – 1999 in their study. We revised the texts, **please see lines 251-252.**

Would you find relevant to reproduce composite differences between Smax and Smin as in e.g. A8 for ERA5 and assess whether the response of temperature and zonal wind in a reanalyzed dataset also reveals a sensitivity to weak and strong solar epochs?

➔ It's hard to assess the sensitivity based on the ERA5 data (1950--present) because it only partly overlaps the strong epoch (i.e., 1932--2014). However, here we reproduced the composite analysis based on the ERA5 (Figure. R2) and discussed the comparison with the modeling results in this study. We should notice the lower top of the ERA5 data (up to 1 hPa) compared to the climate models used in this study. Similar to the modeling results (Figure 4 in manuscript), a weak warm response can be found in the tropical stratosphere in most winter months (except February) with ERA5 data (first row of Figure. R2). However, the stratospheric temperature response in EAR5 does not pass the significant test. Compared with using a short single member, (e.g., EAR5 here), the ensemble mean of climate modeling runs can extract the external forcing signal. The temperature response in FOCI during the strong epoch (second row of Figure A7 in the manuscript) is a bit larger than in the ERA5, which may be due to the warm bias in the tropical upper stratosphere of FOCI (as shown in Figures 10 and 12) leads to a higher sensitivity of the model to the solar forcing. As a result of the increased meridional temperature gradient, response in the zonal mean zonal wind anomalies can be found in the ERA5 (second row of Figure. R2), which is stronger than the FOCI modeling results (second row of Figure. A8) and interrupted by the strong internal variability in February.

[Figure]

**Figure. R2**. Composite differences between solar maximum and minimum of the zonal-mean temperature anomalies (Units: K, first row) and zonal mean zonal wind anomalies (Units: m/s, second row) from ERA5 data (1950--2014).

Using vector figures instead of raster ones may help to improve the quality of your publication.

→ Thanks for the suggestion. We submitted the high-resolution figures (in .pdf formats) to the system for publication.

Due to the extensiveness and unique methodology of the study, I think the whole community would appreciate an adoption of Open Science approaches to allow reproduce the extensive analysis in this study (e.g. Laken, 2016). In particular, I would recommend any kind of willingness of the authors to publish the code allowing to reproduce the figures in the paper. There are multiple ways how to proceed, either to allow the access upon request or via portals allowing to assign Digital Object Identifier (DOI) to the research outputs, e.g. ZENODO. I think it could enhance the quality and reliability of this publication.

→ Thanks for the suggestion. All the codes involved in this study are achievable via the ZENODO link: https://doi.org/10.5281/zenodo.13358940. **We added the description in lines 407-409**.

I really appreciate the authors's willingness to use the robust bootstrap method to but why do you use only 1000 samples? Furthermore, this should be used to assess the significance level of the correlation coefficient to secure methodological consistency. Or was the temporal autocorrelation taken into account in your composites? Can you discuss how the inclusion of the effective sample size (see Section 5 in Bretherton et al, 1999) would influence the t-test results? Do your composite samples comply with the t-test assumptions?

➔ Following the method described by Diaconis and Efron (1983), we performed a 1000-fold bootstrapping test with replacement to estimate the statistical confidence level (90%) of the ensemble mean composites in this study. Here is a bit more explanation of this method and we took a composite of temperature anomalies as an example. (1) We calculated the ensemble mean temperature response to the solar cycle by the difference of ensemble mean temperature anomalies between solar maximum and minimum --- the true value for the bootstrap method. (2) Using all the original data (i.e., all years and all members) as a seeds pool, we calculated an averaged value of the all-years (i.e., 165 years in total) from the seeds pool --- the observed value for a random set. (3) We mimicked step (2) 1000 times by replacing the temperature anomaly randomly. If 90% of the amount of the observed values from step (3) were different from (and smaller than) the true value of step (1), we then marked the true value as a significant response. This method can be used to identify the significance level of the solar signal different from the background noise without assuming that the data have a normal distribution, especially for the cases where only several solar cycles were included (i.e., only a few data).

Of course, we could also mimic the process 10000 times more or increase the critical level (like 95%). We tested the 10000 times and only very tiny changes happened in the results, so we kept the 1000 times to save the computing resources. However, when the critical level is increased to 95%, the significance of the composite of temperature anomalies only reduces a bit but a large reduction in the composite of zonal mean zonal wind anomalies. To facilitate the comparison with our previous work (Drews, et al., 2022) which used the same method, we kept the 90% significance level in this study.

➔ The bootstrap t-test does improve the power of the t-test for a pair of non-normality datasets. In this study, we calculated the correlation coefficients in the 45-year running windows and demonstrated their dependencies on the solar cycle amplitudes. However, the 95% significant levels based on the bootstrap t-test in all the 45-year windows and for all the ensemble members mixed up in one busy figure (in a way as Figure 2), and most of them overlapped. It is hard to interpret and compare. Considering the effective degree of freedom in the 45-year running windows (method described below) are quite similar, we prefer to use a consistent t-value from the two-sides student's t-test to show the 95% significant level (as indicated by the black dash line in Figure 2) to facilitate the comparisons and reproduction. The effective degree of freedom in each 45-year window was calculated following the method used in the work of Pyper and Peterman (1998) and simplified as only the autocorrelation coefficients at lag 1 are considered. More details of this method are also described in the work of Huo et al. (2023).

We briefly described the method of calculating the effective degree of freedom in the method section, **please see lines 143-147.**

**References:**
Diaconis, P. and Efron, B.: Computer-Intensive Methods in Statistics, Scientific American, 248, 116–131, 1983.
Pyper, B. J. and Peterman, R. M.: Comparison of methods to account for autocorrelation in correlation analyses of fish data, Canadian Journal of Fisheries and Aquatic Sciences, 55, 2127–2140, https://doi.org/10.1139/f98-104, 1998.

Huo, W., Xiao, Z., and Zhao, L.: Phase-Locked Impact of the 11-Year Solar Cycle on Tropical Pacific Decadal Variability, Journal of Climate, 36, 421–439, https://doi.org/https://doi.org/10.1175/JCLI-D-21-0595.1, 2023.455

Please specify what CCR in your figures stands for

➔ "CCR." stands for "correlation coefficients in a running window". An explanation is added in the method section. **Please see line 133.**

l288 replace EAR with ERA5

➔ Revised.

l290 replace EAR with ERA5

➔ Revised.

References

Bretherton, C. S., Widmann, M., Dymnikov, V. P., Wallace, J. M., & Bladé, I. (1999). The Effective Number of Spatial Degrees of Freedom of a Time-Varying Field, Journal of Climate, 12(7), 1990-2009. Retrieved Jan 19, 2022, from https://journals.ametsoc.org/view/journals/clim/12/7/1520-0442_1999_012_1990_tenosd_2.0.co_2.xml

Kuchar, A., Sacha, P., Miksovsky, J., & Pišoft, P. (2015). The 11-year solar cycle in current reanalyses: a (non)linear attribution study of the middle atmosphere. Atmospheric Chemistry and Physics, 15(12), 6879–6895. https://doi.org/10.5194/acp-15-6879-2015

Kunze, M., Kruschke, T., Langematz, U., Sinnhuber, M., Reddmann, T., and Matthes, K.: Quantifying uncertainties of climate signals in chemistry climate models related to the 11-year solar cycle – Part 1: Annual mean response in heating rates, temperature, and ozone, Atmos. Chem. Phys., 20, 6991–7019, https://doi.org/https://doi.org/10.5194/acp-20-6991-2020, 2020.

Laken, B. A. (2016). Can Open Science save us from a solar-driven monsoon? Journal of Space Weather and Space Climate, 6, A11. http://doi.org/10.1051/swsc/2016005020.

Milinski, S., Maher, N., and Olonscheck, D.: How large does a large ensemble need to be?, Earth Syst. Dynam., 11, 885–901, https://doi.org/10.5194/esd-11-885-2020, 2020.

Mitchell, D.M., Gray, L.J., Fujiwara, M., Hibino, T., Anstey, J.A., Ebisuzaki, W., Harada, Y., Long, C., Misios, S., Stott, P.A. and Tan, D. (2015), Signatures of naturally induced variability in the atmosphere using multiple reanalysis datasets. Q.J.R. Meteorol. Soc., 141: 2011-2031. https://doi.org/10.1002/qj.2492

Thejll, P., Christiansen, B., and Gleisner, H.: On correlations between the North Atlantic Oscillation, geopotential heights, and geomagnetic activity, Geophys. Res. Lett., 30, https://doi.org/10.1029/2002GL016598, 2003

---

## Author Comment (AC2)

Huo et al. explores a topic of 11-year solar cycle influence on the atmosphere through the top-down mechanism. Surface effects of such mechanism are very uncertain and have been largely disputed in a recent literature, and therefore the authors focus mostly on the middle atmospheric part of the story, where the forcing is created for further potential downward propagation. Authors aim to explore the potential reasons for multi-model uncertainties by looking at the solar signals in the middle atmospheric shortwave heating rates, temperature, ozone, and zonal winds, and by contrasting those to climatological biases of models. This is an interesting topic and the paper also uses a unique set of data (long-term large-ensemble three-model simulations), however it requires some major changes before it can be published.

Here are some major comments (some of them also appear later in the list of specific ones):

1. The paper states itself as an 11-year cycle study, however authors do not focus that much on the max-min differences but mostly on the 45-year means, which is rather representative of the grand minima type of variability.

➔ There is a bit of misunderstanding and we are sorry for the misleading texts. We did not use the 45-year means but calculated the correlation coefficients between a solar forcing index (F10.7 Index in this study) and meteorological variables (e.g., SWHR, temperature) in a "moving" 45-year window.

2. The paper mostly relies on the correlation analysis and seems to not exploit the full potential of the available data. Two out of three experiments are used only for a little bit, even though they are mentioned multiple times throughout the text in descriptions. In fact, I struggled to see where the LOWFREQ one is used at all.

➔ Yes. We mainly used the FULL to investigate the solar signals. It's a pit that we cannot fully use up all the experiments as we expected when we designed them. We tried to include the LOWFREQ when we started this work, like using FULL minus LOWFREQ to reduce the influence of the low-frequency external forcings. However, it introduced an artificial signal, especially for the dynamic response. After discussion, we decided to mainly focus on the FULL and compared the results with FIX at some point to help interpret the results. The texts about LOWFREQ was removed in the revised version.

3. The main findings of the paper are straightforward (higher signals in the lower latitudes and higher altitudes, stronger signals during active phases, model dependent top-down mechanism, and potential causes for it in the model climatological biases), however the way the analysis is created to show this is overcomplicated. There are 12 figures in the main text and 9 in the appendix, while most of them (and related text) show statistically insignificant results or pure empty space, and the text discussing them is way too technical. It looks more like a report rather than a paper aimed at conveying a clear message and findings to the reader. All this can be substantially shortened.

➔ Thanks for the comments. We re-plotted some figures and revised the manuscript accordingly.

4. In the same direction but specifically about the figures: the amount and use of figures needs to be critically reviewed by the authors in a way that figures serve the story and not that the text just describes many similar plots. Also, the choice of Figures in the main

text vs the appendix is sometime confusing. For example, you have almost half of a big paragraph discussing Figure A2, while Figure 7 shows the same as Figure 6 but at a different level and is mentioned in just one sentence. Figures 9 or 11 are just full of empty space.

➔ Revised.

Specific comments:

Title: you analyze only the NH extended wintertime, consider reflecting it in the title somehow

➔ Revised.

l25: "upward" -> "upward-propagating"

➔ Revised.

l35: comma is missing after "The "top-down" mechanism"

➔ Revised.

l50: "much larger than the" -> "much larger than of the"

➔ Revised.

l53: "the dynamics and the uncertainty of the model" - sounds too vague, please rephrase

➔ Revised.

l63 and l65: "upper middle atmosphere" and "lower middle atmosphere" sound odd, as you haven't defined the middle atmosphere boundaries and its lower and upper parts. Consider using "upper stratosphere/lower mesosphere" and "lower stratosphere" instead.

➔ Thanks for the suggestion. The texts were revised.

Section 2.1: It is worth noting that all three models used are of the ECHAM family. There also have been papers intercomparing ECHAM5 and ECHAM6 GCMs, as well as the performances of their (original and modified) radiative transfer schemes, which would be useful in the interpretation of the SWHR and temperature signals.

➔ Thanks for pointing this out. We added a discussion in the revised version. **Please see lines 169-182.**

l93: Explain what is SOLCHEK, it is not described anywhere.

➔ We added a link to the website for the SOLCHECK project in its footnote, though it's in German.

l104: Describe what ozone is used in MPI-ESM-HR and how it is treated in the three experiments

➔ Ozone concentrations from the CMIP6 are used and ozone is treated inactively. A description is added in the revised version. **Please see lines 112-113**.

Section 2.2: You list three experiments together, and it is expected therefore that all of them will be heavily used during the analysis (e.g., as differences between them). However, the FIX and the LOWFREQ ones are used only for a tiny bit. Please highlight that the paper mostly relies on the FULL one and the others are used only for small specific purposes. It is also unclear why you don't look at the classical differences between the experiments and just rely on the correlations of filtered data instead, given also how much statistics you would have with all the ensemble members. FULL-FIX would give you long-term trend + 11-year signal, while FULL-LOWFREQ would give you the 11-year signal, i.e. you would be able to extract both the long-term variability and the 11-year signal without a need for correlations or multi-linear regressions. You are free, of course, to choose what to analyze and what methodology to try, but to me it looks a bit like a missed potential!

➔ Thanks for the suggestion of the method! That's exactly what we planned when we designed the experiments, i.e., we expected to distinguish the "long-term trend+11-year solar signal" by FULL minus FIX, and the "11-year solar signal" by FULL minus LOWFREQ". However, when we performed this strategy of the method on the outputs of each model, it seemed more artificial signal to be introduced than the signal should be extracted. The possible reason is that the ensemble size in this study is not large enough to ensure the low-frequency signal or internal variability is correctly removed. After discussion, we prefer to focus on the FULL experiment to explore/re-examine the signal of the 11-year solar cycle and its stability across the history period based on multiple models. We clarified this point in the revised manuscript. **Please lines 124-125**.

l125: how do you justify a 45-year running window? It basically gives you an average over 4 adjacent solar cycles, but it also greatly decreases the overall signal, given that you average solar min and max years together. This also contradicts the title and the main motivation of the paper, which were stated for the 11-year cyclicity. In your case, for most of the paper you rather explore the long-term variability, i.e. your results are much closer to the grand minima impacts set-up than to the question of how the 11-year cycle modulates the atmosphere, even though the mechanisms are similar.

➔ As we replied above, it seems our texts led to a misunderstanding. We didn't average over the 45-year window but calculated the correlation coefficients between meteorological variables and the solar cycle index (the F10.7 here) in the 45-year running window. Indeed, it reduced the overall signal as only 4 adjacent solar cycles were involved in each window. However, following the method used in the work of Drews et al. (2022) and Chiodo et al. (2019), changes in the correlation coefficient between a meteorological variable (e.g., temperature) and the F1.7 index in the 45-year windows moving across the whole data period, can show us the stability of the relationships somehow. **Please see lines 131-134.**

Drews, A., Huo, W., Matthes, K., Kodera, K., and Kruschke, T.: The Sun's role in decadal climate predictability in the North Atlantic, Atmos. Chem. Phys., 22, 7893–7904, https://doi.org/10.5194/acp-22-7893-2022, 2022.

Chiodo, G., Oehrlein, J., Polvani, L. M., Fyfe, J. C., and Smith, A. K.: Insignificant influence of the 11-year solar cycle on the North Atlantic Oscillation, Nature Geosci, 12, 94–99, https://doi.org/https://doi.org/10.1038/s41561-018-0293-3, 2019

l138: why do you use 90% here and not 95% as for the correlation coefficient?

➔ Although some previous studies show both 90% and 95% significance levels (like Figure 4 of Thiéblemont et al (2015) and Figure 2 of Kuroda, et al., 2022), the solar signal is weak, especially the dynamic responses, only a few regions can pass the 95% significance level. It's a similar situation in our current study, as shown Figure. R1 (below). To facilitate comparisons among models as well as with our previous study (Drews et al., 2022), we use the 90% significance level for the composite.

[Figure]

**Fig. R1.** Composite differences between solar maxima and minima of the ensemble mean zonal-mean zonal wind anomalies (units: ms$^{-1}$) in the FULL experiment for FOCI (top), EMAC (middle), and MPI-ESM-HR (bottom). The 95% significance level for the composite of zonal-mean zonal wind anomalies is indicated by white dots (black hatching) based on a 1000-fold bootstrapping test.

l147: "in the tropical region" - it not only the tropics, but all sunlit regions with the strongest effects in the tropics. Please rephrase, otherwise it reads like the effects are present only in the tropics

➔ Revised. **Please see line 162.**

Figure 1: Please simplify the x-axis title (e.g., why do you need "10.7cm" there?). "F10.7 index (sfu)" or "Solar radio flux (sfu)" would suffice.

➔ Revised. We used the F10.7 index (sfu) as the x-axis title.

L151-152: Why do you use annual mean for SWHR and T but DJF-mean for F10.7?

➔ Thanks for pointing this out. The DJF-mean F10.7 is replaced by the annual mean F10.7 in the revised version.

l159: why do you use 10hPa for ozone? According to all previous studies it maximizes rather around 5 hPa, i.e., between 1 and 10 hPa or 35-40 km depending on a study (Maycock et al., 2018 https://acp.copernicus.org/articles/18/11323/2018/; Ball et al., 2019 https://agupubs.onlinelibrary.wiley.com/doi/full/10.1029/2018GL081501; Dhomse et al., 2022 https://acp.copernicus.org/articles/22/903/2022/)

➔ Yes, the maximum response in the ozone depends on the model used, which is between 7 hPa and 10 hPa in the annual mean ozone mix volume for the three models used in this study. There is no special reason for using the 10 hPa for ozone here. Still, all three models show a significant ozone response at 10 hPa in the annual mean tropical averaged profile (Figure 1), which is more pronounced in the winter months (i.e., reaches maximum at 10 hPa, as shown in Figure 1). So, focusing on 10 hPa for the ozone response makes sense and simplifies the comparison with the dynamical response later on (i.e., zonal wind at 1 hPa and 10 hPa).

l160-165: these periods of non-linearity at f10.7>200 sfu look very interesting. Given that these are not so many data points, why don't you provide the periods specifically? This will simplify potential explanations with other forcings.

➔ This comment is a bit confusing. We are not sure about what the "periods" you mean here because Lines 160-165 refer to the scatter figure (Fig. x in the revised version). If we understood the comment in the right way, there are just 6 years (scatter squares/dots) for F10.7>200 sfu which could be of the solar cycle 19 (i.e., the largest solar cycle in our study period back to 1850). The solar signal will be covered by the large internal variability and inter-annual variability with a very short data period (i.e., 6 years here), and it's hard to find a decadal signal with a statistical method. In our study, we compared the features of the "FULL" experiments with the "FIX" ones to interpret the results.

Figure A2: Please add uncertainty estimates to your lines (either over the ensemble or over the period)

➔ We added the spread over the period (shadow regions) into the figure to indicate the response to different strengths of the solar forcing (solar cycle amplitudes).

l169: Again, here it could be useful to discuss the related performances of the radiative transfer codes (e.g., Sukhodolov et al., 2014 https://gmd.copernicus.org/articles/7/2859/2014/; Nissen et al., 2007 https://acp.copernicus.org/articles/7/5391/2007/ and others)

➔ Thanks for the suggestion and references, we added a discussion in the revised version. **Please see lines 167-182.**

l178-180: It looks like the T-correlation is lower than the SWHR one everywhere and not only during 1850-1920

➔ Sorry for the misleading texts. We re-wrote the sentences. **Please see lines 204-206.**

l192: How can you achieve more than one correlation coefficient between two time-series? What do you mean here?

➔ Sorry for the misleading sentences. We explain it here: There's just one correlation coefficient between the time series of F10.7 and the time series of a meteorology variable (take the SWHR anomaly as an example) in each 45-year window. We got 121 windows running from 1850 to 2014 and hence 121 correlation coefficients in total. Then, we plotted a scatter diagram of the correlation coefficients and the solar cycle amplitudes for all121 windows, as shown in Fig.3. The solar cycle amplitudes are identical in some windows, while the correlation coefficients between the F10.7 and the SWHR anomalies in these windows are different. Therefore, Figure 2 demonstrated that we could have two or more correlation coefficients at an identical solar cycle amplitude for the F10.7 index and SWHR anomaly. We rewrite the sentences to avoid misleading. **Please see lines 217-220**.

Figure 2 and further: not clear what does CCR mean.

➔ Sorry for the missing information. The "CCR" stands for "correlation coefficients in a running window". An explanation is added in the method section. **Please see line 131-133.**

Figure 2 and further: I understand that the relation gets unlinear under high SDs, but how or why do you get two lines over the same SD periods there? Why the shading so weird there and often completely de-attached from the ensemble mean line, while it is stated as ensemble spread, which suggests that the ensemble mean should be somewhere in the middle… Sorry it is a very confusing plot and you need to carefully introduce it to the reader, given that similar plots are used for the rest of the paper

➔ Sorry for the confusing figure, and thanks for pointing it out.  A linear method was used to find the boundaries of the ensemble spread, which failed when more than two boundaries appeared. The figures are updated after we modified the codes.

Large dots in Fig. 3 indicate the correlation of the ensemble mean temperature (also the SWHR and O3) anomalies with the F10.7 index, not the ensemble mean of correlations of single ensemble members with the solar index. As the ensemble mean method can reduce the "noise" of internal variability and therefore extract the solar signal to some extent, we got a higher correlation coefficient between the ensemble mean temperature and the F10.7 than the individual members. This result is similar to the work of Drews et al., 2022 (Fig. 5d as an example). We modified the texts to avoid the misleading. **Please see lines 221-223**.

l197-200: Why do you say that the FOCI correlations are robust if all of them are below your significance threshold? Also, for the other two models most of them are also positive and non-significant, therefore I don't see how you can contrast FOCI to the other two.

➔ Thanks for the good point. We revised the interpretations. **Please see lines 224-231.**

l215-216: I don't see much of a stat-significant warming at tropical 70 hPa.  Also how would you explain stat-significant areas in the troposphere? Is it an artifact of 90% instead of 95%?

➔ No, the warming response at the tropical 70 hPa doesn't pass the significant test (both 90% and 95%) in this study. The stat-significant areas in the troposphere show up for both 90% (Fig. 6 in the manuscript) and 95% (Fig. R2 below) significant levels. We also repeated the test with a 5000-fold bootstrapping, only an ignorable changing happened in the stratosphere. and the stat-significant areas in the troposphere are the same as the current figure. Therefore, it's not an artifact of 90% instead of 95%. But we hard to explain where it come from and the warming is so weak.

[Figure]

**Fig. R2.** The same as Fig. R1, but for the ensemble mean zonal-mean temperature anomalies (Units: K, color shading contours) and the poleward meridional temperature gradients (Units: K(100km) $^{-1}$, contours).

Figure 4: Why the countours get interrupted in the EMAC and MPI-ESM cases?

➔ It's due to the values being too close to 0. We fixed it by releasing the 0-values before plotting.

l217: Note that the 11-year cycle-related lower-strat warming has been heavily disputed (Chiodo et al., 2014 https://acp.copernicus.org/articles/14/5251/2014/; Kuchar et al., 2017 https://agupubs.onlinelibrary.wiley.com/doi/full/10.1002/2017JD026948)

➔ Thanks for the references. We added a short discussion about the second warming response in the lower-stratosphere. **Please see lines 244-250.**

l238: "statistical top-down propagation" – did you mean statistically significant?

➔ Yes. We re-write it.

l250-252: do you associate all stat-significant areas in the troposphere as resulting from the top-down propagation? Or only those around the midlatitude jet?

➔ The change in the midlatitude jet is a direct result of the meridional temperature gradient from the stratosphere to the troposphere. The subtropical jets in the troposphere could be related to the changes in tropospheric circulations.

l251-253: you don't show this, so please mark as "(not shown)"

➔ Revised.

Figure 7: This figure shows the same as Fig 6, but for a different level, almost nothing in it is statistically significant, and you mention it in one sentence only. Given that the figure occupies more than half of the page, it doesn't look worthy to have it in the main text.

➔ It's moved to the supplementary figures.

Figures 8-12: these are big figures with lots of empty space and marginal stat significance. Please find a way to replot what you want to show (signal dependencies on model biases) in a more condensed way. Otherwise it is a waste of space and complication for a reader.

➔ Thanks for the suggestion. We replotted the figures by taking the models' biases as the x-axis.

l274: please specify the period used for ERA5. Also, it doesn't look correct that you compare the ERA5 climatology, shown as one line, to the spread of transient model points and treat it as biases. If you want to show model climatological biases, it will be much clearer to just add a lat-pressure figure to the appendix with models minus ERA5 for T and U. Also, there have been many papers validating these models, so it is necessary to verify if your biases are consistent with those reported in the literature.

➔ We used the period of 1950 to 2014 for ERA5. We compared the models' climatology of temperature (zonal wind) with the ERA5 and added the lat-pressure figures in the supplement. Please see Fig. A7 and Fig. R3 below. Both FOCI and EMAC models have a small warm bias in the upper stratosphere (close to 1 hPa) over the North Pole (Figs. R3a and b). MPI-ESM-HR has a large warm bias in the whole stratosphere from the subtropical to the north Pole region (Fig R3c). EMAC and MPI-ESM-HR have a tropical cold bias, especially in the upper stratosphere (Figs. R3b and c). The stratospheric zonal winds over the polar vortex region are stronger in FOCI than ERA5 (Fig. R3d). Both EMAC and MPI-ESM-HR simulate much weaker stratospheric zonal winds than the ERA5 and therefore much weaker polar night jets in these two models (Fig. R3e and f).

➔ We didn't compare the ERA5 climatology (i.e., an average of 1950-2014) with the transient model points but with the average value of the data in a 45-year window. For example, each dot in Fig. 3 indicates the average temperature in a 45-year window. As the 45-year window moves from 1850 to 2014, we have 121 "climatological" values for 121 windows. In our

manuscript, we investigated the possible impacts of models' biases on the solar signal in the polar night jet anomaly, including the biases in the wind speed averaged over 60-65°N at 1hPa (10hPa), tropical stratopause temperature averaged over 25°S-25°N at 1hPa (10hPa), and the North Pole temperature averaged over 65°N-90°N at 1hPa (10hPa). Here we must notice that the climate states in the 45-year windows could shift among the historical periods (e.g., different temperature climatology of the 45-year window in the early period from in the late period). When we compared all of them with the ERA5 climatology (averaged over 1950-2014), the differences, which we named as model biases, changed in the history period, as shown in Fig. R4 (below). For example, the warm bias of the tropical stratopause in MPI-ESM-HR in the earlier period decreases and turns into cold bias in the late period. As Figs.R3c shows cold bias in the tropical stratopause in MPI-ESM-HR for the common period of 1950-2014, we think the warm bias in the earlier period could be just an overestimation due to the different climate states. EMAC has a large cold bias in the late period (about -6K), which is smaller than 1K before 1900. We add sentences to discuss this point in the manuscript. **Please lines 140-143, lines 320-325, lines 340-342.**

[Figure]

**Fig. R3. First row:** Differences in the zonal mean temperature climatology (K) between models (a. FOCI; b. EMAC; c. MPI-ESM-HR) and ERA5. **Second row:** The same as the first row, but for the zonal mean zonal wind climatology (m/s). The climatology here is defined by the mean value of 1950-2014.

[Figure]

**Fig. R4.** (a.) Differences of the simulated (FOCI (black), EMAC (brown), MPI-ESM-HR (blue)) ensemble-mean zonal-mean zonal wind climatology at 1 hPa (m/s) averaged over the 60-65N and a moving 45-year window from ERA5 climatology of 1950-2014. (b) is the same as (a), but for the tropical stratopause temperature climatology (K), i.e., averaged over 25S-25N and at 1 hPa. (c) is the same as (a), but for the stratopause temperature climatology (K) averaged over the North Pole (65-90N, 1 hPa).

l288 and l290: "EAR5" -> "ERA5"

➔ Revised.

Table 1: "EAR5" -> "ERA5". Also, "December dT" is rather a title for the whole table, while the first-row first-column place should for "level"

➔ Thanks for the suggestion. We modified it.

l290-291: how are you sure that the interactive chemistry is responsible here? Either provide some evidence or rephrase as a potential cause, but better still with references (e.g, Chiodo and Polvani, 2016 https://journals.ametsoc.org/view/journals/clim/29/12/jcli-d-15-0721.1.xml)

➔ We rephrased it because it's hard to find a direct evidence or references for this point.

l295: you say that you use anomalies, however the x-axis units look like it is the absolute values

➔ Sorry for the mistake. It's revised.

l306-307: Doesn't EMAC have the cold bias instead of the warm that you mention?

➔ Yes, it's the cold bias in the tropical stratosphere.

Figure 11: Again, this is just a lot of empty space, that you want to put to the main text

➔ Figure is updated and is moved to the supplement.

l328: "relativley" -> "relatively"

➔ Revised.

l352: Please explain what do you mean by a composite here (i.e., max-min differences)

➔ Yes, it's a composite difference between the solar maxima and minima. We added an explanation in the revised version. Please see lines 388-389.

---

## Author Comment (AC3)

General Comments

This study attempts to ascertain reasons for the diversity in modelling results regarding the 11-year solar cycle signal in the middle atmosphere. It does this by examining the solar signal in three different models, starting with the shortwave heating, ozone, and temperature anomalies at 1hPa (the direct solar signals), and then looking at the temperature and zonal wind anomalies lower down in the stratosphere, to see whether any evidence of a 'top-down' transmission of these initial solar signals can be found. When the models show variation in these indirect solar signals, possible reasons for this are examined.

I think the paper is well and logically structured, the overall scientific treatment is quite rigorous, and the ideas presented relevant to discussions about the atmospheric solar cycle imprint in models. Having said that, I think there are some issues with the paper in its current format which would need to be addressed before publication, outlined below. I would therefore recommend publication with minor revisions.

Specific comments

Please define ECHAM/MESSy (L45)

> **Reply:** The fifth-generation European Centre Hamburg general circulation model (ECHAM5)/Modular Earth Submodel System (MESSy).

Section 2.1 (climate models) needs significant reworking to improve readability. Please define all abbreviations as they appear in the text (e.g. NEMO3.6 L83, JSBACH L83, SOLCHEK L92, T42L47MA L93, MECCA L95, JVAL L94, RAD-FUBRAD L95, UBCNOx L95, GR15L40 L96).

> The descriptions are modified, we added part of the full names for the abbreviations to improve the readability. **Please see lines 85-87, 95-106.**

I'm not sure that figure 1 exactly supports what you say in L151-153. I think you need to add linear trendlines to the figures for values below about 150/180sfu. Also, I think the upper limit of this linear trend is possibly lower than 180sfu, maybe 150sfu. Again, a trendline would help to clarify this.

> **Reply:** The least squared regression lines are added to the Figures. Here, we would like to clarify that we removed the least squares quadratic trend for all the meteorological variables (take SWHR as an example) before we investigated their relationship with the solar cycle. So, the regression lines directly show the SWHR increased when the solar forcing increased (indicated by F10.7 index here).

Figure 1 in general is quite difficult to decipher, given the multiple models plotted and many data points. I would suggest replacing it with line plots, with the lines indicating ensemble-mean values, and a shadow region indicating the ensemble spread, similar to your other figures.

> **Reply:** Thanks for the suggestion. But the scatter plot can directly show some potential relationship between two variables from dimensionless arrays without any "further analysis".

As we explain above, the scatter plots can directly show a tendency of the SWHR anomaly when the strength of the solar forcing increases. We modified the scatter figures. The suggested line plots can be found in Figure A2.

Please explain why the spread in ensemble results above a certain value becomes bifurcated in figures 3, 6,7, 11, as indicated by the shadow regions.

➔ **Reply:** Sorry for the confusion. A linear method was used to find the boundaries of the ensemble spread, which failed when more than two boundaries appeared. The figures are updated.

L192 delete ) after ozone

➔ **Reply:** It's revised.

L196 inset ) after Fig 3

➔ **Reply:** It's revised.

Figures 4 & 5: Upon examination, I am concerned that we are seeing some aliasing with the QBO. Most of the subplots in figure 5 do show definite QBO-like equatorial zonal wind anomalies. FOCI is probably the greatest concern because at least some of these QBO-like zonal wind anomalies appear significant (e.g. Feb-Mar at ~30hPa). This could be significant because the QBO state (easterly vs westerly) has been shown to have an influence on the polar night jet strength, i.e. the Holton-Tan effect (see Holton & Tan 1980). Authors should address this concern, ideally by filtering their results for QBO phase (westerly vs. easterly, or neutral).

➔ **Reply:** Yes, the QBO influences the presentation and propagation of solar signals in the stratosphere. Although both FOCI and MPI-ESM-HR have an internally generated QBO, and thereby, its aliasing with the solar signal is expected to be largely reduced in the ensemble mean composite based on the solar cycle, our study suggests that the QBO signal remains in the average of 9 ensemble members. We added a discussion on this point. **Please see lines 281-291.**

L249 insert such before both: '...such that both the positive...'

➔ **Reply:** The texts are revised.

Figure 11: please adjust the x-axis scale range for each model as the scatter plots are coming out too squished horizontally. Consider doing to same for figures 8 & 9; you can always just highlight the different scale ranges in the figure captions/in-text discussion.

➔ **Reply:** Thanks for the suggestion. We modified the figures.

I think you should consider redoing all your line-style plots that show ensemble members and ensemble means (figures 1-3, 6-12) so they just show the ensemble mean and spread in the ensemble members, like in figure A3.

➔ **Reply:** We re-plotted them.

L264 remove quotation marks around "opposing"

➔ **Reply:** revised.

L274 add ) after Fig. 8

➔ **Reply:** revised.

L287 and L363 change pole to polar

➔ **Reply:** revised.

L288 change EAR5 to ERA5 (also in L290, caption and column heading for table 1)

➔ **Reply:** revised.

References:

Holton, J. R., & Tan, H. C. (1982). The quasi-biennial oscillation in the Northern Hemisphere lower stratosphere. Journal of the Meteorological Society of Japan. Ser. II, 60(1), 140-148.

---

## Author Response (AR2)

Dear Editor,

First of all, Happy New Year, and many thanks for the fast response!

I carefully revised the manuscript and inserted some responses to the comments/suggestions below. All the changes in the manuscript are highlighted in red fonts in the tracking file.

Best wishes,

Wenjuan

- l.270 - rephrase "w.r.t the MPI-ESM-HR contribution ..." with "compared to the MPI-ESM-HR contribution forced with CMIP5 data ...", or similar

➔ Revised.

- l. 131 'stability' -> maybe "reproducibility" or "persistence" would be a better word here?

➔ Thanks for the suggestion. We replaced it with "persistence".

- l. 133 - please spell out / define what CCR stands for (still not fully clear)

➔ Revised, please see lines **131-132**.

- l. 134 - change "reduce the overall signal" with "reduce the overall signal-to-noise ratio compared to the response derived from the full period", or similar

➔ Revised

- correct title formatting in the final manuscript

➔ Corrected.

- l. 322 - "have large uncertainty in the early history periods". First, "history" should be replaced with "historical". Second - it's not clear what exactly do you mean here. Do you mean that the differences in model climatologies for those early periods are small compared to ERA5 (1960-2014), and hence not statistically significant? I find that hard to believe. Or do you mean that if you define those differences this way, these are not really true model 'biases', but rather just reflect a comparison between past and present-day periods? Either way, this should be made clear in the text, please.

➔ Sorry for the confusing description. We rewrote the sentences. Please see lines **320-327**.